# ATEX-CF: Attack-Informed Counterfactual Explanations for Graph Neural Networks

**Yu Zhang**
Aalborg University
Aalborg, Denmark, 9220
yuzhang@cs.aau.dk

**Sean Bin Yang**
Aalborg University
Aalborg, Denmark, 9220
seany@cs.aau.dk

**Arijit Khan**
Bowling Green State University, USA, 43403
Aalborg University, Denmark, 9220
arijitk@bgsu.edu

**Cuneyt Gurcan Akcora**
University of Central Florida
Orlando, FL, USA, 32816
cuneyt.akcora@ucf.edu

## Abstract

Counterfactual explanations offer an intuitive way to interpret graph neural networks (GNNs) by identifying minimal changes that alter a model's prediction, thereby answering "*what must differ for a different outcome?*". In this work, we propose a novel framework, ATEX-CF that unifies adversarial attack techniques with counterfactual explanation generation—a connection made feasible by their shared goal of flipping a node's prediction, yet differing in perturbation strategy: adversarial attacks often rely on edge additions, while counterfactual methods typically use deletions. Unlike traditional approaches that treat explanation and attack separately, our method efficiently integrates both edge additions and deletions, grounded in theory, leveraging adversarial insights to explore impactful counterfactuals. In addition, by jointly optimizing fidelity, sparsity, and plausibility under a constrained perturbation budget, our method produces instance-level explanations that are both informative and realistic. Experiments on synthetic and real-world node classification benchmarks demonstrate that ATEX-CF generates faithful, concise, and plausible explanations, highlighting the effectiveness of integrating adversarial insights into counterfactual reasoning for GNNs. **Our code is available at** **https://github.com/zhangyuo/ATEX_CF**.

## 1 Introduction

Graph neural networks excel at node classification by recursively aggregating neighbor features and graph topology, yet their opaque inference undermines trust in critical applications such as healthcare, finance, and scientific discovery (Chen et al., 2024; Zhong et al., 2025). This limitation has spurred research into GNN explainability, with *counterfactual methods* (Yuan et al., 2023; Qiu et al., 2025; Prado-Romero et al., 2024) in particular aiming to determine the smallest modifications to node features or graph structure that cause a model's prediction to change.

Meanwhile, *adversarial attacks* (Zhang et al., 2024; Zhu et al., 2024; Sun et al., 2023) on GNNs have become an equally important line of research, as GNNs can be undermined by minimal, strategically crafted graph-structure perturbations, highlighting the need for robustness analysis. Consequently, robustness against adversarial attacks has become a key priority in GNN research.

Traditional counterfactual graph generation methods, e.g., CF$^2$ (Tan et al., 2022), GCFExplainer (Huang et al., 2023), primarily rely on *edge deletion* to identify crucial substructures that support a particular prediction. While effective, this deletion-centric perspective overlooks the role of *missing relations* in the original graph whose addition could substantially influence predictions. In parallel, extensive studies in graph adversarial learning have demonstrated that adding a small (e.g., 2) number of carefully selected edges can effectively flip the prediction of a target node (Chen et al., 2025; Zhu et al., 2024). Such added edges—though absent in the input graph—often correspond to semantically plausible and structurally coherent relations.

Despite their importance, current approaches address these two directions largely in isolation. From a counterfactual reasoning perspective, adversarially added edges naturally serve as *actionable candidates* for counterfactual generation: *They represent the minimal structural additions required to alter the model's decision.* However, existing counterfactual methods, which predominantly rely on edge deletion, have largely overlooked the potential of incorporating edge-addition information derived from adversarial attack strategies.

Motivated by these insights, *we design a unified framework, ATEX-CF[1] that incorporates attack semantics into counterfactual generation in a controlled and interpretable manner.* Extending counterfactual generation to include *edge addition* has significant benefits. From a **quantitative** perspective, we demonstrate that edge-addition counterfactuals can (1) increase the likelihood of flipping predictions and (2) achieve this with a smaller perturbation budget. From a **qualitative** perspective, they provide practical advantages: (1) **Complementary explanatory coverage** — while edge-deletion counterfactual identifies which existing relations are crucial for a prediction, edge-addition candidates reveal which missing relations could have altered the outcome. For example, in healthcare, a GNN may classify a patient as low-risk for heart disease due to the lack of an edge representing "symptom–drug correlation", while introducing an edge "patient medication record → cardiac side effects" can flip the prediction and reveal hidden reasoning paths. (2) **Uncovering model bias and data deficiencies** — adding certain edges can divulge over-reliance on specific nodes or structural biases. For example, a paper may be misclassified as "theoretical mathematics" due to missing citation edges to authoritative AI conferences. Introducing an edge "paper → ICLR Best Paper Award" corrects the prediction, highlighting dataset limitations and model vulnerabilities.

**Case Study.** To illustrate the limitations of existing counterfactual methods, consider a scenario from the *Loan-Decision* dataset (Ma et al., 2025). Loan approval is granted when both conditions are met: income $> 5$ and degree $> 3$. Applicant Alice has income 6 (satisfies condition) but degree 3 (fails). The model predicts rejection. Classical **deletion-based** counterfactual methods fail here–removing edges further reduces degree.

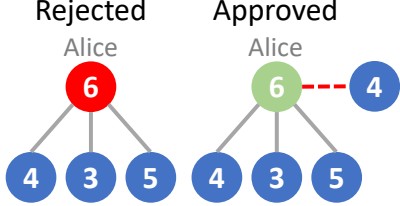

Figure 1: Illustration of counterfactual limitations in the Loan Decision dataset.

Unconstrained **edge additions** (e.g., linking to a billionaire) succeed but can be implausible. Our method ATEX-CF identifies a feasible peer connection that serves as an actionable update and flips the prediction.

While this fusion is promising, combining adversarial attacks with counterfactual explanations is non-trivial(Freiesleben, 2022). Adversarial edges are optimized for misclassifications rather than interpretability, raising challenges in ensuring the qualities of a good counterfactual explanation, such as *high impact*, *sparsity*, and *plausibility* (Longa et al., 2025). Furthermore, when considering missing edge additions to the input graph, the search space of possible perturbations remains combinatorially large, requiring principled mechanisms to balance effectiveness with efficiency.

**Our contributions** can be summarized as follows:

- **Unified perspective.** We establish, for the first time, a theoretical bridge between adversarial attacks and counterfactual explanations in GNNs, showing that adversarial edge additions can be repurposed as counterfactual candidates. This connection provides a principled foundation for unifying attack and explanation.
- **Hybrid counterfactual framework.** We design a novel solution, ATEX-CF, that simultaneously leverages *edge deletions* (traditional counterfactual explanations) and *attack-informed edge additions* (from adversarial strategies), thereby offering a more comprehensive and actionable counterfactual than deletion-only approaches.
- **Enhanced explanatory coverage.** By incorporating edge-addition counterfactuals, ATEX-CF uncovers missing but semantically plausible relations, complements deletion-based explanations, and enables proactive optimization (e.g., suggesting constructive graph modifications rather than only indicating critical existing edges).

---

[1]abbreviation for **At**tack **Ex**planation **C**ounter**f**actual

- **Efficiency and controllability.** We exploit adversarial attack logistics to form a focused candidate space, significantly reducing the combinatorial complexity of our counterfactual search. In addition, ATEX-CF integrates sparsity and plausibility constraints to ensure interpretable and realistic explanations.
- **Empirical validation.** Through experiments on benchmark datasets, we demonstrate that ATEX-CF improves explanatory power, maintains semantic plausibility, and reduces computational burden compared with state-of-the-art counterfactual generation and adversarial attack methods.

## 2 PRELIMINARIES

### 2.1 NODE CLASSIFICATION AND GRAPH NEURAL NETWORKS

**Node Classification in a Graph.** We consider the task of node classification in a graph, denoted as $G = (V, E, \mathbf{X})$, where $V$ is a set of nodes, $E \subseteq \{(v, w) \mid v, w \in V\}$ is a set of undirected, unweighted edges, and $\mathbf{X} = \{\mathbf{x}_0, \mathbf{x}_1, \ldots, \mathbf{x}_{N-1}\}$ comprises node feature vectors with $\mathbf{x}_i \in \mathbb{R}^d$ for each node $v_i$. The adjacency matrix $\mathbf{A} \in \{0, 1\}^{N \times N}$ has entries $\mathbf{A}_{vw} = 1$ if $(v, w) \in E$ and 0 otherwise. A subset $V_L \subseteq V$ is labeled, forming training data; each labeled node has a class $y_v \in \mathcal{C} = \{1, \ldots, c\}$. The goal is to predict the label of a target node $v \in V$ in a supervised manner given $\mathbf{A}$ and $\mathbf{X}$. Key mathematical symbols are summarized in Table 6 in the Appendix.

**Graph Neural Networks.** Graph Neural Networks classify nodes through a message-passing scheme (Kipf & Welling, 2017). Each node representation is iteratively updated by aggregating and transforming information from its neighbors. For the Graph Convolutional Network (GCN), a prominent GNN, the hidden representation at layer $l + 1$ is $\mathbf{H}^{(l+1)} = \sigma\left(\hat{\mathbf{A}}\mathbf{H}^{(l)}\mathbf{W}^{(l)}\right)$, where $\mathbf{H}^{(0)} = \mathbf{X}$, $\sigma$ is a nonlinear activation, $\mathbf{A}_{\text{self}} = \mathbf{A} + \mathbf{I}_N$ augments the adjacency with self-loops, $\mathbf{D}_{ii} = \sum_j (\mathbf{A}_{\text{self}})_{ij}$ is the degree matrix, and $\hat{\mathbf{A}} = \mathbf{D}^{-\frac{1}{2}}\mathbf{A}_{\text{self}}\mathbf{D}^{-\frac{1}{2}}$. The trainable weights at layer $l$ are $\mathbf{W}^{(l)}$. The final output is obtained by applying a softmax to the last hidden layer $\mathbf{Z} = \text{softmax}\left(\hat{\mathbf{A}}\mathbf{H}^{(K)}\mathbf{W}^{(K)}\right)$, with $\mathbf{Z} \in \mathbb{R}^{N \times c}$ giving class probability distributions. Row $\mathbf{Z}_v$ is the distribution for node $v$, and the predicted class is $\hat{y}_v = \arg\max \mathbf{Z}_v$.

### 2.2 GNN EXPLANATIONS

GNN explanation methods (Yuan et al., 2023; Longa et al., 2025) reveal the structural and feature-based evidence that plays a key role in predictions. We categorize them into two paradigms:

**Factual explanations** identify subgraphs or features *supporting* the original prediction. For a target node $v$, an explanation subgraph $G_v \subseteq G$ satisfies $f(G_v, \mathbf{X_v}) = f(G, \mathbf{X_v})$, where $f$ is the GNN model and $\mathbf{X_v}$ denotes the features of $v$. The GNNExplainer method (Ying et al., 2019) optimizes $G_v$ to maximize mutual information with the prediction.

**Counterfactual explanations** identify *minimal perturbations* $\Delta\mathbf{A}$ to alter target node $v$'s prediction $f(\mathbf{A}, \mathbf{X}, v) \neq f(\mathbf{A} \odot \Delta\mathbf{A}, \mathbf{X}, v)$, s.t. $\|\Delta\mathbf{A}\|_0 \leq \kappa$, where $\kappa$ is a perturbation budget.

### 2.3 ADVERSARIAL ATTACKS ON GNNS

Adversarial attacks deliberately perturb graphs (including edge-based and feature-based perturbation) to mislead predictions. Key categories include i) evasion and ii) poisoning attacks.

**Evasion attacks** modify the graph *during inference* without retraining. For target node $v$, edge-based attackers solve $\max_{\Delta\mathbf{A}} \mathcal{L}(f(\mathbf{A} \odot \Delta\mathbf{A}, \mathbf{X}, v), y_v)$ s.t. $\|\Delta\mathbf{A}\|_0 \leq \kappa$, where $\mathcal{L}$ is the loss function which quantifies prediction error.

**Poisoning attacks** corrupt the *training graph* to degrade retrained models. For target node $v$, edge-based attackers optimize $\max_{\Delta\mathbf{A}} \mathcal{L}(f_{\theta^*}(\mathbf{A}, \mathbf{X}, v), y_v)$ s.t. $\theta^* = \arg\min_\theta \mathcal{L}(f_\theta(\mathbf{A} \odot \Delta\mathbf{A}, \mathbf{X}))$, $\|\Delta\mathbf{A}\|_0 \leq \kappa$.

Table 1 summarizes GNN explanations and adversarial attacks according to edge-based perturbation methods by their core characteristics.

Table 1: Comparison of GNN explanation and attack paradigms. We use $E^-$ to denote edge deletions (removing existing edges) and $E^+$ to denote edge additions (introducing new edges).

| Category | Goal | Primary Operation | Example |
|---|---|---|---|
| Factual Expl. | Explain prediction | Identify key subgraph | GNNExplainer (Ying et al., 2019) |
| Counterfactual Expl. | Alter prediction | Mainly $E^-$ (edge deletions), though some recent work includes $E^+$ | CF-GNNExplainer (Lucic et al., 2022) |
| Evasion Attack | Misclassify node | $E^+/E^-$ in inference, often $E^+$ dominant | TDGIA (Zou et al., 2021) |
| Poisoning Attack | Degrade model | $E^+/E^-$ in training, often $E^+$ dominant | Nettack (Zügner et al., 2018) |

**Key Insight.** While counterfactual explanations have historically emphasized $E^-$ to reveal model fragility, adversarial attacks often exploit $E^+$ by introducing new connections. More importantly, the attack literature has developed efficient methods to select which edges to add/delete under small perturbation budgets (e.g., $\kappa = 1, \ldots, 5$), despite the combinatorially large number of possible additions in graphs, making naive counterfactual search impractical. This potential synergy between counterfactual reasoning and attack strategies motivates our problem formulation (§2.4) and the unified framework we propose in (§4).

In this work, we focus on structural evasion attacks, which directly modify the graph topology at inference time. These attacks are particularly suited to our counterfactual framework, as they avoid retraining and yield interpretable perturbations that align with our goals of sparsity and plausibility.

## 2.4 Problem Formulation

Given a graph $G = (V, E, \mathbf{X})$ with adjacency matrix $\mathbf{A}$ and node features $\mathbf{X}$, and a pre-trained GNN classifier $f$, our goal for a target node $v \in V$ is to find a small set of edge perturbations $\Delta\mathbf{E} = \Delta\mathbf{E}^+ \cup \Delta\mathbf{E}^-$, corresponding to additions ($\Delta\mathbf{E}^+$) and deletions ($\Delta\mathbf{E}^-$), such that the prediction for $v$ flips while the resulting counterfactual graph remains *interpretable* and *plausible*. This problem combines two perspectives: from the attack literature, where efficient methods have been developed to select high-impact edge additions under small budgets, and from counterfactual explanations, where minimal and semantically meaningful deletions expose decision-supporting edges. We formalize this hybrid objective in §4.

## 3 A Dual Approach of Explanations and Attacks for GNNs

We develop a theoretical framework that links targeted structural evasion attacks on graph neural networks with instance-level counterfactual explanation subgraphs. The core objective is to formalize when and why adversarial perturbations can serve as building blocks for counterfactual explanations. To this end, we introduce a hypothesis to capture the relationship between the attack subgraph and the counterfactual explanation of a target node. **More importantly, we support this hypothesis with gradient-based reasoning and empirical similarity measures in Appendix A.12.** To the best of our knowledge, this hypothesis and supporting evidence are presented for the first time as a plausible and empirically grounded link between adversarial perturbations and counterfactual explanations in graph learning.

To compare explanation and attack subgraphs, we consider two forms of graph similarity: i) structural similarity (Doan et al., 2021): overlap in nodes or edges, measurable via graph edit distance, and maximum common subgraph metrics. ii) semantic similarity (Bai et al., 2020): closeness in learned graph-level embeddings, indicating similar functional or predictive roles even if the structures differ.

Hypothesis 1 states that the added edges in a successful evasion attack overlap with the most influential edges in a pre-attack counterfactual explanation subgraph.

**Hypothesis 1.** *For a target node $v$, let $\Delta G(E^+)$ denote the set of added edges in an evasion attack that flips the prediction of $f$, and let $CFEx(G)$ denote the pre-attack counterfactual explanation subgraph of the graph $G$. Then, there exists a high graph similarity between $\Delta G(E^+)$ and $CFEx(G)$. The proof is provided in the Appendix A.12.1.*

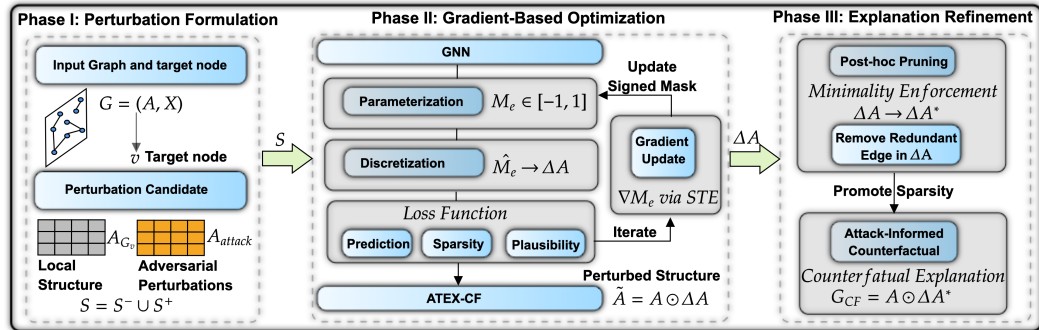

Figure 2: End-to-end workflow of the ATEX-CF framework for counterfactual edge generation.

Building on the hypothesis, in Appendix A.12, we also present two propositions and two corollaries that formalize when attack-based additions outperform deletions in flipping GNN predictions. These results characterize conditions under which deletions provably fail, yet targeted additions succeed, focusing on the functional advantage of attack-informed counterfactuals.

## 4  ATEX-CF: METHODOLOGY FOR COUNTERFACTUAL GENERATION

Our objective is to design a counterfactual explainer that simultaneously incorporates **edge addition** ($E^+$) and **edge deletion** ($E^-$), combining GNN adversarial attacks with counterfactual explanation concepts. This explainer should generate high-impact perturbations while maintaining interpretability and realism. In particular, we jointly optimize three core objectives: **Impact** — efficacy in altering model predictions; **Sparsity** — minimal edits for interpretability; **Plausibility** — semantic validity of graph modifications. Figure 2 illustrates the end-to-end architecture of our ATEX-CF framework, which unifies adversarial edge perturbations with counterfactual explanation generation through a joint optimization of impact, sparsity, and plausibility.

To operationalize these objectives, we cast counterfactual generation as an optimization problem over edge perturbations. Given a candidate set $\mathcal{S}$ of feasible edge edits, we search for $\Delta\mathbf{A} \in \mathcal{S}$ that flips the prediction of the target node while balancing sparsity and plausibility. This is achieved by defining a composite loss with three components, corresponding to our objectives.

**Loss Function.** We formulate counterfactual generation as

$$\min_{\Delta\mathbf{A}\in\mathcal{S}} \mathcal{L}(\Delta\mathbf{A}) = \lambda_1 \mathcal{L}_{pred}(\Delta\mathbf{A}) + \lambda_2 \mathcal{L}_{dist}(\Delta\mathbf{A}) + \lambda_3 \mathcal{L}_{plau}(\Delta\mathbf{A}), \tag{1}$$

where $\mathcal{S}$ is the candidate search space. Here $\mathcal{L}_{pred}$ enforces label flipping, $\mathcal{L}_{dist}$ penalizes the number of edge edits, and $\mathcal{L}_{plau}$ enforces plausibility constraints. Weights $\lambda_i \geq 0$ balance these terms. Next, we will define each loss function.

**Prediction Loss.** We denote by $f(\mathbf{A}_v, \mathbf{X}_v; \mathbf{W})$ the prediction for node $v$ under the original adjacency $\mathbf{A}_v$, and by $g(\mathbf{A}_v, \mathbf{X}_v, \mathbf{W}; \Delta\mathbf{A})$ the prediction under a perturbed adjacency $\mathbf{A}_v \odot \Delta\mathbf{A}$. Both share the same weights $\mathbf{W}$; the difference lies only in the perturbation $\Delta\mathbf{A}$.

To encourage prediction flips, we define the loss as

$$\mathcal{L}_{pred}(\Delta\mathbf{A}) = -\mathbb{I}[f(\mathbf{A}_v, \mathbf{X}_v; \mathbf{W}) = f(\mathbf{A}_v \odot \Delta\mathbf{A}, \mathbf{X}_v; \mathbf{W})] \\ \cdot \mathcal{L}_{\text{NLL}}(f(\mathbf{A}_v, \mathbf{X}_v; \mathbf{W}), g(\mathbf{A}_v, \mathbf{X}_v, \mathbf{W}; \Delta\mathbf{A})). \tag{2}$$

The indicator ensures that the loss is active only when the perturbed graph yields the same prediction as the original. In that case, the negative log-likelihood term penalizes the perturbed prediction, pushing it away from the original class. Once a flip occurs, the loss becomes zero. Although this objective is non-differentiable due to the discrete nature of the indicator function, we employ the straight-through estimator (STE) to enable gradient-based optimization, as detailed in §4.2.

**Sparsity Loss.** To encourage concise and interpretable modifications, we impose a sparsity penalty on the number of structural edits. Specifically, we minimize the $\ell_0$ norm of the adjacency change

$\Delta \mathbf{A} = \Delta \mathbf{E}^+ \cup \Delta \mathbf{E}^-$, where $\Delta \mathbf{E}^+$ and $\Delta \mathbf{E}^-$ denote the sets of added and removed edges, respectively. $\mathcal{L}_{dist}(\Delta \mathbf{A}) = \|\Delta \mathbf{A}\|_0$.

The objective $\mathcal{L}_{dist}(\Delta \mathbf{A})$ measures the total number of edits. By requiring $\|\Delta \mathbf{A}\|_0$ to be small, we keep the modified graph close to the original, curb unnecessary complexity, and reduce overfitting.

**Plausibility Loss.** When generating counterfactual graphs by adding/removing edges, we must control the plausibility of the produced structure. For example, in a citation graph, an old article cannot cite a more recent article. The plausibility penalty discourages unnatural degree/motif changes: $\mathcal{L}_{plau}(\Delta \mathbf{A}) = \mathcal{C}(\Delta \mathbf{A}) = \alpha_{deg} \cdot \mathrm{DegAnom}(\Delta \mathbf{A}) + \alpha_{motif} \cdot \mathrm{MotifViol}(\Delta \mathbf{A})$. We tune $\alpha_{deg}$ and $\alpha_{motif}$ to enforce realism; larger $\alpha_{deg}$ avoids implausible degree jumps, larger $\alpha_{motif}$ avoids implausible clustering jumps.

$$\mathrm{DegAnom}(\Delta \mathbf{A}) = \sum_{v_i \in V_{\mathrm{sub}}} \frac{\left| \deg_{\tilde{\mathbf{A}}_{v_i}}(v_i) - \deg_{\mathbf{A}_{v_i}}(v_i) \right|}{1 + \deg_{\mathbf{A}_{v_i}}(v_i)}, \tag{3}$$

$$\mathrm{MotifViol}(\Delta \mathbf{A}) = \sum_{v_i \in V_{\mathrm{sub}}} \left| c_{\tilde{\mathbf{A}}_{v_i}}(v_i) - c_{\mathbf{A}_{v_i}}(v_i) \right|. \tag{4}$$

DegAnom penalizes large relative changes in node degree to prevent structural anomalies, where $\deg_{\mathbf{A}}(v)$ and $\deg_{\tilde{\mathbf{A}}}(v)$ are degrees of node $v$ before and after modification. MotifViol penalizes drastic changes in local motifs, measured via clustering coefficients $c_{\mathbf{A}_v}(v)$ and $c_{\tilde{\mathbf{A}}_v}(v)$.

## 4.1 CANDIDATE SELECTION

As a key aspect in ATEX-CF, we constrain the search space of possible perturbations $\Delta \mathbf{A}$ to a pre-selected candidate set $\mathcal{S}$. This tractable set is constructed through a dual mechanism that incorporates both **local neighborhood structures** and **non-local, attack-informed candidates**, balancing interpretability with the ability to discover impactful counterfactuals.

**Edge Deletion Candidates ($\mathcal{S}^-$):** We follow the principle of *actionability* and *plausibility* (Wachter et al., 2017); counterfactual explanations should suggest meaningful changes within an entity's sphere of influence (e.g., local graph neighborhood), rather than involving arbitrary, distant entities. As a result, candidate edges for removal are restricted to the existing edges within the $(l+1)$-hop neighborhood $\mathcal{N}^{l+1}(v)$ of the target node $v$, i.e., $\mathcal{S}^- = \{e \mid e \in E, e \in \mathcal{N}^{l+1}(v)\}$.

**Edge Addition Candidates ($\mathcal{S}^+$):** To overcome the limitation of deletion-only approaches and incorporate insights from adversarial attacks, our key innovation is to draw candidate edges for addition from adversarial attack subgraphs. Specifically, we employ the latest GOTTACK method (Alom et al., 2025) to generate a set of candidate edges $\Delta A_{\mathrm{attack}}$ for the target node $v$. GOTTACK identifies influential nodes for edge addition by learning the **graph orbit characteristics** of nodes that, when connected to $v$, maximally increase the probability of misclassification. An orbit in graph theory represents the role of a node within its local substructure (e.g., a central node in a star graph). The underlying Hypothesis 1 of GOttack, validated by our experiments in Table 18, is that nodes occupying similar structural roles (orbits) often have similar predictive influences on the target node. Therefore, edges suggested by GOTTACK (connecting $v$ to nodes in specific, influential orbits) are both highly impactful and structurally coherent.

**Final Candidate Set and Local Graph Formation:** The complete candidate set is the union $\mathcal{S} = \mathcal{S}^- \cup \mathcal{S}^+$. The adjacency matrix $\mathbf{A}_v$ for the local subgraph used in subsequent optimization (Eq. 2) is then formed by combining the original $(\ell+1)$-hop neighborhood structure of $v$ and the adversarial perturbation candidates:

$$\mathbf{A}_v = \underbrace{\mathbf{A}_{G_v}}_{\text{local structure}} + \underbrace{\Delta \mathbf{A}_{\mathrm{attack}}}_{\text{adversarial perturbations}} \tag{5}$$

This formulation provides a focused and principled search space $\mathcal{S}$ that is crucial for the efficiency and effectiveness of our counterfactual search algorithm. We use the $(l+1)$-hop neighborhood because an $l$-layer GCN aggregates information from nodes up to $l$ hops away; including the $(l+1)$-hop ensures that all nodes and edges within the target's effective receptive field—including those that can indirectly influence its representation—are considered as candidates.

## 4.2 SIGNED-MASK PERTURBATION AND FORWARD DISCRETIZATION

After candidate edges are selected, the challenge is to optimize over the discrete choices of additions and deletions. Since direct optimization of binary graph structures is non-differentiable, we employ a continuous signed mask relaxation. In the forward pass, the mask is discretized into $\{-1, 0, +1\}$ to yield concrete perturbations, while in backpropagation, the straight-through estimator treats this step as identity, allowing gradients to propagate through discrete edge decisions. This process is carried out as follows and the complete ATEX-CF framework is given in Alg. 1.

Each candidate edge $e \in \mathcal{S}$ (where $\mathcal{S}$ is the candidate set defined in §4.1) is associated with a continuous signed parameter $M_e \in [-1, 1]$ (Line 1 of Alg. 1). This parameter encodes both the directionality and the magnitude of the proposed modification; a signed mask variable $M_e$ encodes perturbations, with $M_e > 0$ denoting an edge addition ($e \in \Delta \mathbf{E}^+$), $M_e < 0$ denoting an edge deletion ($e \in \Delta \mathbf{E}^-$), and $M_e \approx 0$ no modification. Here, the sign of $M_e$ indicates the type of operation (addition or deletion), while the magnitude $|M_e|$ reflects the proposed strength or importance of the perturbation. This continuous representation facilitates gradient-based learning.

During the forward pass, we discretize these continuous parameters to obtain a binary perturbation matrix. This process involves two steps: thresholding and sparsity enforcement. First, we apply thresholding to convert $M_e$ into a ternary value. The discretized mask is obtained by thresholding, $\widehat{M}_e = +1$ if $M_e > \tau^+$, $\widehat{M}_e = -1$ if $M_e < -\tau^-$, and $\widehat{M}_e = 0$ otherwise (Line 3 of Alg. 1).

where $\tau^+$ and $\tau^-$ are positive thresholds that control the sensitivity for edge addition and deletion, respectively. Typically, we set $\tau^+ = \tau^- = 0.5$ to ensure symmetry.

Next, to enforce the perturbation budget constraint $\|\Delta \mathbf{A}\|_0 \leq \kappa$, we retain only the $\kappa$ edges with the largest magnitudes $|M_e|$ and assign their discretized values $\widehat{M}_e \in \{-1, 0, +1\}$ to the corresponding entries in the adjacency matrix (Line 4 of Alg. 1). The perturbation matrix is defined as $\Delta \mathbf{A}_{i,j} = \widehat{M}_e$ if edge $(i, j)$ is among the top-$\kappa$ candidates ranked by $|M_e|$, and 0 otherwise. This ensures that at most $\kappa$ edges are modified, producing sparse and interpretable counterfactuals.

The objective loss and resulting perturbed adjacency matrix is then computed as $\widetilde{\mathbf{A}} = \mathbf{A} \odot \Delta \mathbf{A}$, where the operator $\odot$ applies the signed edge modifications encoded in

---

**Algorithm 1: ATEX-CF: Counterfactual Generator**

**Require:** Graph $G = (\mathbf{A}, X)$, model $f$, target node $v$, candidate set $\mathcal{S}$
1: Initialize mask $M_e \leftarrow \mathbf{0}$ over $\mathcal{S}$
2: **for** $t = 1$ to $T_{\max}$ **do**
3: $\quad \widehat{M}_e \leftarrow$ THRESHOLD$(M_e, \tau^+, \tau^-)$ $\quad\triangleright$ Discretize
4: $\quad \Delta \mathbf{A} \leftarrow$ TOP-$\kappa(|M_e|)$ $\quad\triangleright$ Sparsify
5: $\quad$ Evaluate $\mathcal{L}(M_e)$ on $\mathbf{A} \odot \Delta \mathbf{A}$
6: $\quad M \leftarrow M - \eta \nabla_M \mathcal{L}(M)$ $\quad\triangleright$ Update via STE
7: $\quad$ **if** flipped($v$) **and** $\|\Delta \mathbf{A}\|_0$ stable **then**
8: $\quad\quad$ **break**
9: $\quad$ **end if**
10: **end for**
11: **return** PRUNE$(\Delta \mathbf{A}, G, f, v)$ $\quad\triangleright$ See Alg. 2

---

$\widehat{M}_e \in \{-1, 0, +1\}$ (Line 5 of Alg. 1). To maintain differentiability through this discretization step, we employ the straight-through gradient estimator (STE) during backpropagation $\frac{\partial \widehat{M}_e}{\partial M_e} \approx 1$. This approximation (Line 6 of Alg. 1) allows gradients to flow directly through the binarization operation, treating the discretization as if it were an identity function in the backward pass (Bengio et al., 2013). Consequently, the continuous parameters $M_e$ can be updated using gradient descent, even though the forward pass involves non-differentiable operations. This approach is widely used in training binary neural networks and has been shown to be effective in practice. The loop stops early if the target node $v$ flips and the perturbed edges $\|\Delta \mathbf{A}\|_0$ are stable (Line 7 of Alg. 1).

**Minimality-Aware Post-Hoc Pruning** While the training loss promotes sparsity and plausibility in expectation, the discrete relaxation can leave redundant edges active in $\Delta \mathbf{A}$. This occurs mostly due to noisy or approximate gradient updates that over-compensate. To enforce the minimality of counterfactual explanations, we adopt a simple yet effective greedy algorithm (Alg. 2 in Appendix A.4). Edges in the candidate set are ranked by their importance score $\psi_e \propto |\partial \mathcal{L}/\partial M_e|$ (approximated gradient magnitude at line 2 of Alg. 2). The algorithm then iteratively removes the least important edge, checking if the prediction flip persists (Line 3 of Alg. 2). This continues until no more edges can be removed without reverting the prediction, and it attains final perturbation $\Delta \mathbf{A}^*$.

## 5 EXPERIMENTS

### 5.1 EXPERIMENTAL SETUP

Table 2: Dataset statistics.

| Dataset | Homophily Ratio | #Nodes | #Edges | #Features | #Classes | Type |
|---|---|---|---|---|---|---|
| BA-SHAPES (Ying et al., 2019) | 0.80 | 700 | 3958 | – | 4 | Synthetic |
| TREE-CYCLES (Ying et al., 2019) | 0.90 | 871 | 1,940 | – | 2 | Synthetic |
| Loan-Decision (Ma et al., 2025) | 0.47 | 1000 | 3950 | 2 | 2 | Synthetic |
| Cora (Sen et al., 2008) | 0.81 | 2,708 | 5,429 | 1,433 | 7 | Real |
| Chameleon (Pei et al., 2020) | 0.24 | 2,277 | 36,101 | 2,325 | 5 | Real |
| Ogbn-Arxiv (Hu et al., 2020) | 0.66 | 169,343 | 1,166,243 | 128 | 40 | Real |

**Datasets.** We evaluate ATEX-CF on both synthetic and real-world benchmarks. Synthetic datasets include **BA-SHAPES** and **TREE-CYCLES** (Ying et al., 2019), widely used in GNN explainability, and the **Loan-Decision** social graph (Ma et al., 2025). For real-world evaluation, we use the **Cora** citation network (Sen et al., 2008) and the large-scale **ogbn-arxiv** dataset from OGB (Hu et al., 2020). Additionally, we include the heterophilic **Chameleon** dataset (Rozemberczki et al., 2021), which is known for its low feature homophily and non-community structure, providing a challenging real-world setting for counterfactual explanations.

**GNNs.** We evaluate our approach on three standard GNN architectures: **GCN** (Kipf & Welling, 2017), **GAT** (Velickovic et al., 2018), and **Graph Transformer** (Shi et al., 2021).

**Baselines**: We compare our method against a comprehensive set of baseline approaches, which we categorize into two groups. The first group comprises **explanation-based baselines**: **CF-GNNExplainer** (Lucic et al., 2022), **CF$^2$** (Tan et al., 2022) and **NSEG** (Cai et al., 2025), counterfactual methods that optimize for edge deletions using a perturbation mask; **INDUCE** (Verma et al., 2024), an inductive counterfactual framework that learns structural interventions through local subgraph rewiring, and can naturally realize both edge deletions and edge additions; **C2Explainer** (Ma et al., 2025), a customizable mask-based counterfactual explainer that jointly optimizes edge and feature perturbations under flexible constraints, supporting both removal and insertion of edges. **GNNExplainer** (Ying et al., 2019), a factual explainer adapted for counterfactual analysis by removing edges in descending order of importance until prediction flips; and **PGExplainer** (Luo et al., 2020), another factual method adapted similarly to GNNExplainer. The second group consists of **attack-based baselines** repurposed for counterfactual generation: **Nettack** (Zügner et al., 2018), a white-box adversarial attack method adapted by using its edge perturbation capability such that the target class is different from the original prediction; and **GOttack** (Alom et al., 2025), a recent adversarial method that leverages graph orbital theory to identify critical nodes for edge additions, making it naturally suited for generating addition-based counterfactuals. For fair comparison, all methods are constrained to a default perturbation budget (i.e., maximum possible number of edge flips) of $\kappa = 5$ edges. We vary $\kappa$ for ablation study in Figure 3. Explanation-based methods (CF-GNNExplainer, GNNExplainer, PGExplainer) are restricted to edge deletions only, while attack-based methods (Nettack, GOttack) and our ATEX-CF can use both edge additions and deletions within the same budget.

In our experiments, we ran each experiment with three different random seeds (102, 103, and 104), and report the mean and standard deviation across runs. For the attack model, we employed evasion attacks using the GOttack method. For the ATEX-CF, we used a learning rate of 0.001, trained for 200 epochs, and adopted the SGD optimizer to generate counterfactual explanation with a maximal perturbed budget of 5 edges. The default loss weights were configured as follows: $\lambda_1 = 1.5$, $\lambda_2 = 0.5$, $\lambda_3 = 0.5$, $\alpha_{deg} = 1.5$, and $\alpha_{motif} = 1.0$. These hyperparameters were chosen to balance prediction flipping, sparsity, and plausibility in counterfactual generation.

**Evaluation Metrics**: We evaluate the performance of counterfactual explainers in misclassification rate, fidelity, explanation size, plausibility, and time costs. Definitions are given in Appendix A.5.

Table 3: Meta Results. Average ranks (↓) across six datasets (lower is better). Ranks are computed per metric per dataset (best=1; ties get the same rank), then averaged across datasets equally. "Wins" counts how many times a method achieved rank one across all metric–dataset cells (6 datasets ×5 metrics = 30 cells, ties allowed).

| Method | Misclass. | Fidelity | $\Delta$E | Plausibility | Time (sec) | Overall Avg. | Wins |
|---|---|---|---|---|---|---|---|
| CF-GNNExplainer | 4.7 | 4.8 | 2.0 | 2.3 | 9.5 | 4.67±2.68 | 1 |
| INDUCE | 6.3 | 6.8 | 4.5 | 7.8 | 2.8 | 5.67±1.78 | 1 |
| C2Explainer | 5.0 | 6.2 | 5.7 | 5.8 | 8.7 | 6.27±1.26 | 1 |
| CF$^2$ | 7.0 | 7.0 | 6.7 | 5.7 | 7.8 | 6.83±0.70 | 0 |
| NSEG | 6.7 | 6.7 | 7.5 | 4.3 | 6.7 | 6.37±1.07 | 0 |
| GNNExplainer | 6.8 | 7.5 | 4.7 | 5.5 | 3.3 | 5.57±1.49 | 1 |
| PGExplainer | 8.2 | 7.7 | 4.2 | 5.8 | **1.0** | 5.37±2.60 | 6 |
| Nettack | 3.3 | 2.5 | 8.8 | 8.0 | 4.5 | 5.43±2.53 | 2 |
| GOttack | 4.8 | 4.3 | 8.8 | 8.0 | 3.0 | 5.80±2.23 | 0 |
| ATEX-CF (ours) | **1.2** | **1.3** | **1.0** | **1.2** | 7.3 | **2.40±2.47** | **20** |

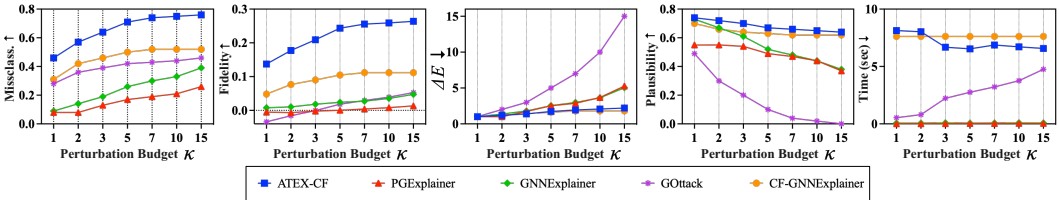

Figure 3: Counterfactual explanations on **Cora** and GCN under varying perturbation budgets $\kappa$

## 5.2 Results and Analysis

We evaluate ATEX-CF against all baselines under the same budget constraints ($\kappa = \{1, \dots, 5\}$). Table 3 summarizes average rankings across datasets and metrics. Our method achieves the best overall rank (2.40 vs. 4.67 for the next best) and wins 20/30 metric–dataset combinations, far exceeding competitors. This confirms that ATEX-CF consistently finds more effective counterfactuals. In particular, counterfactual explainers (CF-GNNExplainer, CF$^2$, NSEG, GNNExplainer, PGExplainer) are limited to edge deletions, while counterfactual explainers (INDUCE, C2Explainer), adversarial attack methods (Nettack, GOttack) and ATEX-CF can also add edges. Crucially, CF-GNNExplainer explicitly seeks minimal edge deletions, and GOttack systematically manipulates graph orbits to induce errors, yet neither matches ATEX-CF on combined effectiveness and realism. **Our empirical results on individual datasets and with other GNNs (GAT and Graph Transformer) are given in Appendix A.6–A.7.**

Figure 3 plots performance vs. perturbation budget on Cora. As the budget grows, all methods improve: ATEX-CF quickly raises misclassification (e.g., from 0.46 at $\kappa$=1 to 0.76 at $\kappa$=15) far above others, and maintains the highest fidelity and plausibility. Notably, ATEX-CF 's edit size increases only mildly with $\kappa$, whereas attack baselines must exhaust all allowed edits ($\Delta E \rightarrow 5$). This trend illustrates that our objective effectively exploits additional budget to find better counterfactuals without excessive edits.

**Ablation Study.** Table 15 in Appendix A.9 shows the effect of removing each loss. Our findings demonstrate that $\mathcal{L}_{dist}$ enforces concise edits, $\mathcal{L}_{plau}$ preserves semantic plausibility, and their combination in ATEX-CF achieves the best overall balance across all metrics.

**Sensitivity Analysis.** We next analyze key hyperparameters. **Search depth** ($l$): Figure 4 in Appendix A.10 shows that $l = 2$ captures sufficient local structure surrounding the target node for effective counterfactuals. **Hyperparameters** ($\alpha_{deg}, \alpha_{motif}$): Figure 5 in Appendix A.10 demonstrates that ATEX-CF is robust across a range of hyperparameter values (e.g., $\alpha = 0.5$–1.5); while moderate $\alpha$ maximizes fidelity and plausibility together.

**Impact of Pruning Strategy.** We also evaluate the impact of our candidate-edge pruning strategy (Algorithm 2 in Appendix A.4). As shown in Figure 6, pruning yields more concise explanations by reducing redundant edge edits ($\Delta \mathbf{A} = 1.71 \rightarrow 1.62$), but as intended, its real utility is the reduced

runtime (6.12s $\rightarrow$ 3.00s), while preserving predictive accuracy (misclass.=0.71), plausibility (0.76 vs. 0.75), making it an effective and efficient enhancement of our framework.

## 5.3 ASYMMETRIC COSTS OF EDGE PERTURBATIONS

**Feasibility of Edits.** While our framework assumes that edge additions and deletions are possible, not all structural edits are equally realistic in every domain. For example, in citation networks, adding an edge between two papers may be implausible after publication. To handle this, we incorporate a real-valued cost metric into our optimization that allows domain-specific constraints on edit feasibility.

The default perturbation budget in ATEX-CF assumes symmetric costs for edge additions and deletions, treating both equally under the constraint $\|\Delta \mathbf{A}\|_0 = \|\Delta E^+\| + \|\Delta E^-\| \leq \kappa$. To account for scenarios where additions may be less actionable or more expensive (e.g., in real-world graphs with immutable structures), we introduce a scalar weight $C > 0$ to control the cost asymmetry between perturbation types. The new constraint becomes:

$$C \cdot \|\Delta E^+\| + \|\Delta E^-\| \leq \kappa.$$

We vary $C$ over a range of values (from 0.5 to 21.0) to analyze how penalizing edge additions affects counterfactual quality. Table 4 reports results on Cora using a GCN under budget setting: $\kappa = 20$. More results under different $\kappa$ are provided in Appendix A.13.

Table 4: Counterfactual performance under asymmetric addition cost $C$ with $\kappa = 20$.

| Addition Cost | Deletion Cost | Misclass.↑ | Fidelity↑ | $\Delta E$ (E$^+$, E$^-$)↓ | Plausibility↑ | Time (sec)↓ |
|---|---|---|---|---|---|---|
| 0.5 | 1.0 | **0.70** | **0.23** | 1.78 (0.78, 1.00) | **0.72** | **6.1** |
| 0.8 | 1.0 | **0.70** | **0.23** | 1.78 (0.77, 1.01) | 0.71 | **6.1** |
| 1.0 | 1.0 | **0.70** | **0.23** | 1.78 (0.77, 1.01) | 0.71 | 10.3 |
| 3.0 | 1.0 | **0.70** | **0.23** | 1.82 (0.66, 1.16) | 0.69 | 10.7 |
| 5.0 | 1.0 | **0.70** | **0.23** | 1.82 (0.65, 1.17) | 0.69 | 10.9 |
| 10 | 1.0 | 0.69 | **0.23** | 1.80 (0.61, 1.19) | 0.69 | 11.0 |
| 15 | 1.0 | 0.69 | **0.23** | 1.76 (0.60, 1.16) | 0.69 | 11.0 |
| 20 | 1.0 | 0.54 | 0.15 | **1.42 (0.49, 0.93)** | 0.68 | 11.5 |
| 21 | 1.0 | 0.42 | 0.10 | 1.78 (0.00, 1.78) | 0.62 | 11.7 |

As the addition cost $C$ increases relative to the deletion cost, the optimizer favors deletion-heavy or deletion-only solutions. This leads to a trade-off: higher $C$ reduces misclassification success and fidelity, and in extreme cases ($C \geq 20$), restricts exploration to only deletions, thereby weakening plausibility. These results confirm that asymmetric weighting serves as an effective control knob for tailoring explanation strategies in cost-sensitive domains, but also warn against overly skewed values that impair explanation quality.

## 6 CONCLUSIONS

We presented ATEX-CF, a theoretically grounded framework that unifies adversarial attacks and counterfactual explanations for graph neural networks. By incorporating both edge additions and deletions under a constrained budget, ATEX-CF generates explanations that are not only faithful but also informative. Our joint optimization of fidelity, sparsity, and plausibility ensures instance-level counterfactuals that balance interpretability with realism. Experiments on synthetic and real-world benchmarks confirm the effectiveness of this integration, highlighting how adversarial insights can substantially improve the quality of counterfactual explanations, compared with state-of-the-art counterfactual generation and adversarial attack methods. Future directions include extending ATEX-CF to dynamic graphs, incorporating node feature perturbations, and applying the framework to real-world domains like healthcare and finance.

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
