REPRODUCIBILITY STATEMENT

We provide the full implementation of our models and experimental setup to ensure reproducibility. Experimental results are reported as the mean and standard deviation across different random seeds, and the hyperparameters used are detailed in Section 5.1. **Our code and data are available at `https://github.com/zhangyuo/ATEX_CF`.**

## A APPENDIX

### A.1 LIMITATION

A key limitation of this study is the assumption that edge additions and deletions are equally feasible, which may not hold in domains where graph modifications are inherently constrained. Future work could incorporate domain-specific constraints and node-feature perturbations to enhance the practical relevance of ATEX-CF while preserving its theoretical contributions. The central premise of our approach is that unifying adversarial attack strategies with counterfactual reasoning strengthens both the fidelity and plausibility of explanations. Unlike methods that treat these perspectives independently, ATEX-CF provides a principled integration that balances model sensitivity with explanation realism in a computationally tractable way.

Our current framework does not consider poisoning attacks or node feature perturbations. Poisoning requires retraining the model after each perturbation, which conflicts with our fixed-model counterfactual setup. Feature perturbations, while useful in some domains, are harder to constrain plausibly and often lack structural interpretability in graph settings. Extending ATEX-CF to incorporate these forms of attacks is a promising direction for future work.

We also recognize that the current plausibility loss, based on structural proxies such as degree anomaly and motif violations, may only partially address the risk of generating out-of-distribution explanations. These generic regularizers do not account for domain-specific constraints such as temporal consistency or semantic incompatibility (e.g., citing future papers or linking unrelated functional modules). To mitigate this, we emphasize that our pruning step (Algorithm 2) serves as a late filter that discards non-essential edits, including those that may be structurally valid yet semantically implausible. In practice, pruning significantly reduces the number of active perturbations, helping ensure that final explanations are not only minimal but more likely to remain within the distribution.

### A.2 RELATED WORK

We begin a complete account of counterfactual explanations and adversarial examples with their philosophical origins, where both have been studied as forms of contrastive reasoning and causal dependence.

(Freiesleben, 2022) examines the connection between counterfactual explanations and adversarial examples in standard machine learning, studying their shared optimization goal and conceptual and historical development in fields such as philosophy and psychology. The analysis remains focused on models with independent input features, where perturbations do not propagate through structured dependencies. Our work builds on this by studying the connection in graph neural networks, where node predictions depend on message passing over edges. In this setting, identifying minimal changes that flip predictions becomes harder, and naive perturbations can easily break plausibility. We address these challenges by using adversarial edge additions to guide counterfactual generation and show that attack-informed edits offer an effective and realistic way to produce explanations in graphs.

**GNN Explanations.** Different categories of GNN explanation methods have been developed to offer diverse perspectives and improve the interpretability of GNN models (Khan & Mobaraki, 2023; Yuan et al., 2023). Two main categories of explanations persist: factual and counterfactual. **Counterfactual explanations**, which are the focus of this work, provide explanations by identifying the minimum perturbation or change to the input graph that leads to a different prediction from the model (Bajaj et al., 2021; Huang et al., 2023; Tan et al., 2022), thereby revealing the most critical structures underlying the decision. Existing methods are predominantly based on edge deletions. For

instance, CF-GNNExplainer (Lucic et al., 2022), RCExplainer (Bajaj et al., 2021), GNN-MOExp (Dandl et al., 2020), CF$^2$ (Tan et al., 2022), NSEG (Cai et al., 2025), Banzhaf (Chhablani et al., 2024), and CF-GFNExplainer (He et al., 2024) all design deletion-oriented mechanisms, such as gradient-based mask optimization, decision boundary constraints, multi-objective optimization, or probabilistic sampling. These approaches emphasize faithfulness, sparsity, or necessity/sufficiency guarantees, but rely mainly on removing salient substructures.

More recently, several works on node classification have extended counterfactual explanations to include edge additions, or the joint use of both addition and deletion. INDUCE (Verma et al., 2024) treats counterfactual search as a Markov decision process, allowing the model to learn edge modifications (both additions and deletions) that lead to flips. C2Explainer (Ma et al., 2025) further integrates hypergraph representations with straight-through optimization to balance reliability and fidelity, and explicitly models the potential risks of false evidence from edge additions. In the context of graph classification, approaches such as counterfactual graphs (Abrate & Bonchi, 2021), CLEAR (Ma et al., 2022), GCFExplainer (Kosan et al., 2025), and density-based counterfactual graphs (Abrate et al., 2023) adopt generative or global search strategies that combine edge addition and deletion to ensure causally consistent and semantically coherent explanations.

Overall, while edge-deletion-based methods dominate current counterfactual explanation research, the emerging edge-addition or mixed approaches demonstrate that edge addition can serve as a complementary mechanism, especially in cases where deletion-based explanations fail to capture counterfactual reasoning. This motivates our design of hybrid counterfactual explainers that leverage both deletion and addition. Unlike prior counterfactual methods that may include edge additions, our approach is the first to integrate adversarial attack strategies—systematically leveraging their capacity to identify high-impact edge additions—with traditional deletion-based reasoning, thereby unifying two separately studied domains to generate more effective and actionable explanations. Table 5 summarizes important GNN explanation methods, including both factual and counterfactual approaches, along with their explanation type, candidate modification, and target task.

Table 5: Characteristics important GNN explainers including ours. "E , F, N" denote removing/adding edges, node feature modification, removing/adding nodes, respectively. GC and NC denote graph classification and node classification, respectively.

| Method | Type | Candidate | Task |
|---|---|---|---|
| GNNExplainer (Ying et al., 2019) | factual/instance-level | E, N | GC/NC |
| PGExplainer (Luo et al., 2020) | factual/instance-level | E | GC/NC |
| MOO (Liu et al., 2021) | counterfactual/instance-level | E(-), N(-) | NC |
| CF-GNNExplainer (Lucic et al., 2022) | counterfactual/instance-level | E(-) | NC |
| RCExplainer (Bajaj et al., 2021) | counterfactual/instance-level | E(-) | GC/NC |
| CF$^2$ (Tan et al., 2022) | counterfactual/instance-level | E(-), F | GC/NC |
| INDUCE (Verma et al., 2024) | counterfactual/instance-level | E(+,-) | NC |
| NSEG (Cai et al., 2025) | counterfactual/instance-level | E(-), F | GC/NC |
| Banzhaf (Chhablani et al., 2024) | counterfactual/instance-level | E(-) | NC |
| C2Explainer (Ma et al., 2025) | counterfactual/instance-level | E(+,-), F | GC/NC |
| ATEX-CF (ours) | counterfactual/instance-level | E(+,-) | NC |

**GNN Adversarial Attacks.** Graph adversarial attacks investigate structural perturbations but from a different perspective: their objective is to reduce model performance rather than to improve interpretability. These attacks can be divided into two main categories: evasion attacks and poisoning attacks (Yuan et al., 2023; Longa et al., 2025). In evasion attacks, the GNN parameters are fixed and the adversary perturbs the test graph to flip predictions without retraining. Examples include targeted edge modifications during inference (Zou et al., 2021; Chang et al., 2020; Ma et al., 2020; Fan et al., 2023). Poisoning attacks, in contrast, manipulate the training data by injecting adversarial samples, forcing the retrained model to internalize the perturbations and degrade performance (Alom et al., 2025; Zügner et al., 2018; Li et al., 2023; Chen et al., 2018; Geisler et al., 2021).

Empirical studies show that adversarial evasion attacks on GNNs — particularly those based on strategically adding edges — exploit data biases and model weaknesses to induce misclassifications, in stark contrast to counterfactual explanations, which predominantly rely on edge deletions. Inte-

grating these attack-inspired edge-addition perturbations into counterfactual frameworks can enrich explanation graphs and forge a novel link between adversarial robustness and interpretability.

**Pitfalls in Explanations** Faber et al.(Faber et al., 2021) identify five pitfalls that affect the evaluation of GNN explanation methods: (1) the GNN may rely on bias terms or spurious features rather than the annotated evidence, (2) the ground-truth explanation may be redundant or non-unique, leading to mismatches during scoring, (3) some datasets allow trivial explanations, such as those based on nearest neighbors or centrality, (4) weak models that do not learn the true structure render all explanation assessments unreliable, and (5) explanation behavior may vary significantly across architectures even with similar accuracy. These concerns are valid for attribution-based explanations evaluated against fixed motifs, but they do not apply to our setting. Our method generates counterfactuals by identifying sparse edge perturbations that flip the model's prediction. We do not assume or require ground-truth substructures, nor do we compare explanations to predefined motifs. Instead, our evaluation reflects the model's actual behavior and is based on plausibility and edit sparsity. This makes our approach robust to the ground-truth mismatches discussed in(Faber et al., 2021; Agarwal et al., 2022).

**Fusing GNN Explanations and Robustness against Attacks.** Recent efforts on robust explainable graph neural networks combine explainability with adversarial defense to preserve explanation quality under worst-case perturbations. GNNEF (Li et al., 2024) reveals that perturbation-based explainers (e.g., GNNExplainer, PGExplainer) are highly fragile, as minor structural changes can drastically alter explanations without affecting predictions, and proposes loss- and deduction-based attacks exposing this vulnerability across both graph- and node/edge-level tasks. (Fan et al., 2023) develop GEAttack that can attack both a GNN model and its explanations by simultaneously exploiting their vulnerabilities. (Chanda et al., 2025) exploit explainability-based strategy to devise adversarial attacks on GNNs. Complementarily, (Lukyanov et al., 2025) introduce a benchmark analyzing the interplay between robustness and interpretability under poisoning and evasion attacks, showing that most defenses improve interpretability but with architecture-dependent trade-offs and limitations in existing metrics. Building on these insights, XGNNCert (Li et al., 2025) provides the first certifiable robustness guarantee for graph-level tasks, ensuring stable explanations without sacrificing predictive performance. At the node/edge level, $k$-RCW (Qiu et al., 2024) proposes robust counterfactual witnesses (RCWs) that remain factual, counterfactual, and resilient to structural disturbances, while GNNNIDS (Galli et al., 2025) introduces an evaluation framework for intrusion detection via structural adversarial attacks, demonstrating that Integrated Gradients produces precise yet exploitable explanations. While these works improve the robustness of explanations, to the best of our knowledge, we are the first to unify adversarial attack techniques such as both edge additions and deletions for better counterfactual explanation generation.

### A.3 SUMMARY OF NOTATIONS USED IN THIS PAPER

Table 6 provides a concise summary of the key notations used in this paper, covering graph structure, node features, GNN models, optimization terms, and theoretical concepts.

### A.4 MINIMALITY-AWARE POST-HOC PRUNING: ALGORITHM 2

After the main optimization phase completes, the resulting perturbation $\Delta \mathbf{A}$ may contain extraneous edges introduced by approximation errors in gradient updates or overly cautious thresholding. To refine the explanation and ensure that every retained edge meaningfully contributes to the prediction flip, we apply a post-hoc minimality-aware pruning procedure, formalized in Algorithm 2. This algorithm iteratively ranks edges in $\Delta \mathbf{A}$ by their estimated importance (measured via gradient magnitude) and removes them greedily in ascending order of this score. An edge is retained only if its removal would reverse the counterfactual prediction, ensuring that the final perturbation $\Delta \mathbf{A}^*$ is irreducible by construction.

This refinement step improves the explanation's conciseness without requiring retraining or re-optimization. In practice, we observe that pruning reduces the average number of edits from 1.71 to 1.62, while maintaining the same level of predictive fidelity and plausibility. It also yields a significant runtime improvement (6.12s to 3.00s) due to the smaller edge set being evaluated downstream

Table 6: Summary of notations used in this paper

| Symbol Group | Description |
|---|---|
| **Graph Structure** | |
| $G, V, E$ | Input graph, node set, edge set |
| $N, m$ | Number of nodes and edges ($N = |V|$, $m = |E|$) |
| $\mathbf{A}, \mathbf{A}_{self}$ | Adjacency matrix $\mathbf{A} \in \{0,1\}^{n \times n}$, adjacency matrix with self-loops ($\mathbf{A} + \mathbf{I}_N$) |
| $\mathbf{D}, \hat{\mathbf{A}}$ | Degree matrix, normalized adjacency matrix ($\mathbf{D}^{-\frac{1}{2}} \mathbf{A}_{self} \mathbf{D}^{-\frac{1}{2}}$) |
| $\widetilde{\mathbf{A}}, \Delta\mathbf{A}$ | Perturbed adjacency matrix ($\mathbf{A} \odot \Delta\mathbf{A}$), edge modifications ($\in \{-1, 0, 1\}^{n \times n}$) |
| $\Delta\mathbf{E}^+, \Delta\mathbf{E}^-$ | Added/deleted edge sets |
| **Node Features, Neighborhood, & Labels** | |
| $\mathcal{N}^l(v)$ | $l$-hop neighborhood of node $v$ |
| $\mathbf{X}$ | Node feature matrix ($\in \mathbb{R}^{n \times d}$) |
| $v, y_v, \hat{y}_v$ | Target node, ground-truth label, predicted label |
| **GNN Model** | |
| $\mathbf{W}^{(l)}, \mathbf{H}^{(l)}$ | Weight matrix and hidden representations at GNN layer $l$ |
| $\mathbf{Z}$ | Output logits |
| $f(\mathbf{A}, \mathbf{X}, v)$ | GNN prediction for node $v$ |
| **Optimization & Loss** | |
| $\mathcal{L}(\bullet)$ | Loss objective function |
| $\mathcal{L}_{pred}, \mathcal{L}_{dist}$ | Prediction loss (flipping), sparsity loss (minimal edits) |
| $\mathcal{L}_{plau}, \mathcal{C}(\Delta\mathbf{A})$ | Plausibility loss, plausibility penalty |
| $\|\Delta\mathbf{A}\|_0, \kappa$ | Number of changed edges, perturbation budget |
| $M_e, \widehat{M}_e$ | Continuous signed mask ($\in [-1, 1]$), discretized mask ($\in \{-1, 0, 1\}$) |
| $\tau^+, \tau^-$ | Positive/negative thresholds for discretization |
| $\psi_e, \mathcal{S}$ | Edge importance score, candidate modification set |
| $\lambda_1, \lambda_2, \lambda_3$ | Loss trade-off weights |
| $\alpha_{deg}, \alpha_{motif}$ | Realism penalty weights |
| $\eta$ | Learning rate |
| **Theoretical Concepts** | |
| $m_v$ | Prediction margin |
| $g_e$ | Gradient influence: $\frac{\partial m_v}{\partial A_e}$ |
| $\mathbf{A}_v, \mathbf{D}_v, \tilde{\mathbf{D}}_v$ | Local adjacency, degree matrix, perturbed degree matrix for node $v$ |
| $c_{\mathbf{A}}(v), c_{\widetilde{\mathbf{A}}}(v)$ | Clustering coefficient (original/perturbed) for node $v$ |

(Figure 6). Thus, pruning serves both an interpretability function by enforcing parsimony, and a practical one by improving efficiency.

## A.5 EVALUATION METRICS

- **Misclassification Rate**: It measures the fraction of predictions flipped by perturbations. Higher values indicate stronger disruption, consistent with the *attack success rate* widely used in GNN adversarial attacks, such as Nettack (Zügner et al., 2018) and GOttack (Alom et al., 2025).

$$\text{Misclassification Rate} = \frac{1}{N} \sum_{i=1}^{N} \mathbb{I}(\hat{y}_i^{1-m_i} \neq c_i), \qquad (6)$$

where $N$ is the number of evaluated target nodes, $c_i = f(\mathbf{A}, \mathbf{X}, v_i)$ denotes the model-predicted class of target node $v_i$. $\hat{y}_i^{1-m_i} = f(\tilde{\mathbf{A}}, \mathbf{X}, v_i)$, $\tilde{\mathbf{A}}$ is the perturbed adjacency, $m_i$ is the explanation mask (edges added/removed), $\mathbb{I}$ is the indicator function.

---

**Algorithm 2** Minimality Pruning

---

**Input:** Perturbation $\Delta\mathbf{A}$, graph $G$, model $f$, target node $v$
**Output:** Minimal perturbation $\Delta\mathbf{A}^*$

1. Initialize: $\Delta\mathbf{A}^* \leftarrow \Delta\mathbf{A}$
2. Rank edges in $\Delta\mathbf{A}^*$ by importance score $\psi_e$ (descending)
3. **for each** edge $e_i$ in ascending order of $\psi_e$:
   (a) $\Delta\mathbf{A}' \leftarrow \Delta\mathbf{A}^* \setminus \{e_i\}$ (tentatively remove)
   (b) **if** $f(\mathbf{A} \odot \Delta\mathbf{A}', v) \neq f(\mathbf{A}, v)$:
      i. $\Delta\mathbf{A}^* \leftarrow \Delta\mathbf{A}'$ (keep the smaller perturbation)
4. **return** $\Delta\mathbf{A}^*$

---

- **Fidelity**: This metric measures the prediction confidence drop on the model's predicted class $c_i$ (Bajaj et al., 2021). Formally:

$$\text{Fidelity} = \tfrac{1}{N} \sum_{i=1}^{N} \left( f(\mathbf{A}, \mathbf{X}, v_i)_{c_i} - f(\tilde{\mathbf{A}}, \mathbf{X}, v_i)_{c_i} \right), \tag{7}$$

where $f(\mathbf{A}, \mathbf{X}, v)_c$ denotes the softmax probability assigned to class $c$. Unlike the binary Misclassification Rate, which captures label flips, Fidelity provides a finer-grained sensitivity analysis by quantifying how perturbations reduce the model's confidence in its own prediction.

- **Explanation Size $\Delta\mathbf{E}$**: It represents the average number of structural modifications (including both edge additions and deletions) made per counterfactual explanation, calculated as:

$$\Delta\mathbf{E} = \tfrac{1}{n} \sum_{i=1}^{n} \Delta\mathbf{E_i} = \tfrac{1}{n} \sum_{i=1}^{n} (\Delta\mathbf{E_i}^+ + \Delta\mathbf{E_i}^-), \tag{8}$$

We report the average over successful counterfactuals $n$ since $\Delta\mathbf{E}_i$ is well-defined only when a valid counterfactual is generated, ensuring that the metric reflects the true complexity of feasible explanations rather than being diluted by failed cases (Lucic et al., 2022; Tan et al., 2022). Here, $\Delta\mathbf{E_i}$ represents the set of perturbed edges for node $v_i$. Smaller values indicate more compact and interpretable explanations.

- **Plausibility**: This evaluates the human-interpretable quality of counterfactual explanations by assessing their realism and coherence with domain knowledge. The plausibility score is averaged across $n$ successful counterfactuals:

$$\text{Plausibility} = \tfrac{1}{n} \sum_{i=1}^{n} S_{plau}^{(i)}, \quad S_{plau}^{(i)} = 2 \cdot \left( 1 - \tfrac{1}{1+\exp(-k \cdot L_{plau}^{(i)})} \right), \tag{9}$$

where $S_{plau}^{(i)} \in (0,1)$ is the plausibility score for target node $v_i$, $k$ is a scaling factor (default $k = 1$), and $L_{plau}^{(i)} \in (0, \infty)$ encodes domain-specific constraints quantifying the realism of the counterfactual. Higher values indicate more plausible explanations. In our experiments, $L_{plau}^{(i)}$ is instantiated using the definition in Eq.3 and Eq.4 in §4, ensuring consistency with our evaluation setup. More generally, $L_{plau}^{(i)}$ serves as a flexible placeholder that can incorporate task-specific structural and semantic constraints to assess the realism of counterfactuals in diverse domains.

- **Time Cost**: We record the average running time required in seconds to generate a counterfactual explanation for a single node, providing insights into the computational efficiency of different methods.

## A.6 EXPERIMENTAL RESULTS ON INDIVIDUAL DATASETS

Across all datasets, ATEX-CF flips the most target nodes while using very few edits. On Cora (Table 7), ATEX-CF achieves a misclassification rate of 0.72 with only 1.63 average edge changes, compared to only 0.53 for both Nettack and GOttack (each being forced to flip 5 edges). Our fidelity (0.2336) and plausibility (0.75) are also the highest. In contrast, PGExplainer is extremely fast (0.04s) but flips almost no nodes, and attack methods (Nettack/GOttack) flip all 5 edges but

Table 7: Performance of counterfactual explanations on **Cora** and GCN.

| Method | Base GNN | Misclass. ↑ | Fidelity ↑ | $\Delta E(E^+, E^-)$ ↓ | Plausibility ↑ | Time (sec) ↓ |
|---|---|---|---|---|---|---|
| CF-GNNExplainer | GCN | 0.49±0.013 | 0.1060±0.0034 | 1.70±0.08 (0.00, 1.70) | 0.64±0.008 | 10.21±2.88 |
| INDUCE | GCN | 0.17±0.008 | 0.0256±0.0014 | 2.66±0.04 (1.50, 1.16) | 0.27±0.008 | 0.88±0.08 |
| C2Explainer | GCN | 0.61±0.029 | 0.2116±0.0046 | 1.95±0.03 (0.58, 1.37) | 0.68±0.008 | 8.25±0.12 |
| CF$^2$ | GCN | 0.40±0.012 | 0.0813±0.0077 | 2.66±0.07 (0.00, 2.66) | 0.61±0.016 | 7.08±0.20 |
| NSEG | GCN | 0.33±0.012 | 0.0815±0.0077 | 3.26±0.15 (0.00, 3.26) | 0.58±0.008 | 3.47±0.14 |
| GNNExplainer | GCN | 0.22±0.016 | 0.0197±0.0150 | 2.58±0.13 (0.00, 2.58) | 0.53±0.021 | 0.44±0.52 |
| PGExplainer | GCN | 0.14±0.009 | -0.0010±0.0017 | 2.38±0.03 (0.00, 2.38) | 0.53±0.005 | **0.04±0.02** |
| **Attack Models** | | | | | | |
| Nettack | GCN | 0.53±0.005 | 0.1484±0.0057 | 5.00±0.00 (3.86, 1.14) | 0.13±0.005 | 3.36±0.85 |
| GOttack | GCN | 0.53±0.005 | 0.1466±0.0043 | 5.00±0.00 (4.70, 0.30) | 0.10±0.000 | 2.24±0.81 |
| ATEX-CF (Ours) | GCN | **0.72±0.008** | **0.2336±0.0003** | **1.63±0.01 (0.90, 0.73)** | **0.75±0.008** | 7.26±2.5 |

Table 8: Performance of counterfactual explanations on **BA-SHAPES** and GCN.

| Method | Base GNN | Misclass. ↑ | Fidelity ↑ | $\Delta E(E^+, E^-)$ ↓ | Plausibility ↑ | Time (sec) ↓ |
|---|---|---|---|---|---|---|
| **Explainers** | | | | | | |
| CF-GNNExplainer | GCN | 0.64±0.017 | 0.3383±0.0079 | 1.33±0.20 (0.00, 1.33) | 0.57±0.012 | 11.30±3.72 |
| INDUCE | GCN | 0.70±0.005 | 0.3475±0.0140 | 1.64±0.02 (0.40, 1.24) | 0.28±0.008 | 0.25±0.02 |
| C2Explainer | GCN | 0.59±0.008 | 0.1426±0.0058 | 3.79±0.01 (0.02, 3.77) | 0.08±0.005 | 1.55±0.02 |
| CF$^2$ | GCN | 0.58±0.012 | 0.1264±0.0052 | 2.87±0.05 (0.00, 2.87) | 0.07±0.008 | 5.22±0.17 |
| NSEG | GCN | 0.19±0.005 | 0.0269±0.0009 | 3.36±0.10 (0.00, 3.36) | 0.67±0.008 | 2.71±0.09 |
| GNNExplainer | GCN | 0.65±0.022 | 0.3055±0.0019 | 1.83±0.15 (0.00, 1.83) | 0.34±0.009 | 0.81±1.03 |
| PGExplainer | GCN | 0.73±0.031 | 0.3672±0.0015 | 1.45±0.05 (0.00, 1.45) | 0.41±0.075 | **0.03±0.01** |
| **Attack Models** | | | | | | |
| Nettack | GCN | 0.64±0.005 | 0.3526±0.0063 | 5.00±0.00 (4.08, 0.92) | 0.22±0.0036 | 0.89±0.38 |
| GOttack | GCN | 0.63±0.012 | 0.3399±0.0081 | 5.00±0.00 (4.30, 0.70) | 0.32±0.008 | 0.73±0.22 |
| ATEX-CF (Ours) | GCN | **0.83±0.009** | **0.4237±0.0118** | **1.24±0.02 (1.21, 0.03)** | **0.71±0.000** | 8.96±0.43 |

yield very low plausibility ($\approx 0.1$–$0.13$). A similar pattern holds on BA-Shapes (Table 8) and Tree-Cycles (Table 9), which are motif-based synthetic graphs. On BA-Shapes, ATEX-CF attains 0.83 misclassification with $\Delta E$=1.24 and plausibility 0.71, clearly outperforming others; on Tree-Cycles, it achieves 0.58 misclassification vs. 0.74 for C2Explainer, but with far higher plausibility (0.64 vs. 0.14) and much smaller edits ($\Delta E$=1.29 vs. 3.09). These synthetic benchmarks have no node features and explicit motif structures, and ATEX-CF reliably discovers the minimal motif changes needed.

On the Loan-Decision social graph (Table 10), ATEX-CF again dominates: 0.68 misclassification (vs. $\leq 0.45$ for others) and highest fidelity (0.3658) with only $\Delta E$=1.27. Finally, on the large real ogbn-arxiv network (Table 11), ATEX-CF flips 0.90 fraction of nodes vs. 0.85–0.88 for C2Explainer and attacks, yet uses just $\Delta E$=1.20 edges (C2Explainer uses 1.62 and attacks use 5) and achieves plausibility 0.73 (vs. 0.58–0.70). The ogbn-arxiv dataset is a citation graph of CS papers with 128-dimensional features and 40 classes, confirming ATEX-CF scales to large, feature-rich graphs.

Finally, to further assess generalization beyond homophilic benchmarks, we include experiments on the heterophilic Chameleon dataset (Table 14). Chameleon poses additional challenges due to feature heterophily and non-community structure. Nonetheless, ATEX-CF achieves the highest misclassification rate (0.84) and fidelity (0.2595) while requiring only 1.64 edits—substantially fewer than other baselines, which require more edits (vs. 1.81-5). ATEX-CF also maintains high plausibility (0.76), comparable to the best-performing explainers yet with significantly stronger effectiveness.

These results confirm that ATEX-CF remains robust across both homophilic and heterophilic graphs. In summary, our method consistently finds compact counterfactual edits that flip more predictions than baselines, yielding higher fidelity while preserving realistic graph structure.

## A.7    EXPERIMENTAL RESULTS WITH GRAPH TRANSFORMER AND GAT

Tables 12 and 13 further demonstrate that ATEX-CF remains consistently superior on both Graph Transformer (Shi et al., 2021) and GAT (Velickovic et al., 2018) backbones. On Graph Transformer (Table 12), our method achieves the highest misclassification rate (0.44) and plausibility (0.50), while also maintaining competitive edit compactness ($\Delta E = 1.66$). On GAT (Table 13), ATEX-CF shows an even clearer margin, boosting misclassification to 0.47 and plausibility to 0.65,

Table 9: Performance of counterfactual explanations on **TREE-CYCLES** and GCN.

| Method | Base GNN | Misclass. ↑ | Fidelity ↑ | $\Delta \mathbf{E}(\mathbf{E}^+, \mathbf{E}^-)$ ↓ | Plausibility ↑ | Time (sec) ↓ |
|---|---|---|---|---|---|---|
| **Explainers** | | | | | | |
| CF-GNNExplainer | GCN | 0.49±0.054 | 0.3437±0.0422 | 1.95±0.03 (0.00, 1.95) | 0.34±0.005 | 6.16±2.09 |
| INDUCE | GCN | 0.53±0.005 | 0.3194±0.0065 | 2.76±0.01 (0.89, 1.87) | 0.22±0.009 | **0.01±0.00** |
| C2Explainer | GCN | **0.74±0.005** | 0.2783±0.0116 | 3.09±0.07 (0.13, 2.96) | 0.14±0.005 | 10.15±0.32 |
| CF$^2$ | GCN | 0.46±0.019 | 0.2531±0.0128 | 3.83±0.05 (0.00, 3.83) | 0.31±0.012 | 9.16±0.07 |
| NSEG | GCN | 0.45±0.012 | 0.2175±0.0094 | 3.76±0.07 (0.00, 3.76) | 0.29±0.012 | 5.34±0.20 |
| GNNExplainer | GCN | 0.53±0.085 | 0.3608±0.0637 | 2.57±0.31 (0.00, 2.57) | 0.26±0.041 | 0.70±0.93 |
| PGExplainer | GCN | 0.41±0.033 | 0.2733±0.0288 | 2.52±0.12 (0.00, 2.52) | 0.31±0.022 | **0.01±0.00** |
| **Attack Models** | | | | | | |
| Nettack | GCN | 0.58±0.022 | **0.4508±0.0217** | 5.00±0.00 (4.34, 0.66) | 0.27±0.099 | 0.58±0.17 |
| GOttack | GCN | 0.18±0.005 | 0.1083±0.0033 | 5.00±0.00 (4.91, 0.09) | 0.21±0.016 | 0.41±0.09 |
| ATEX-CF (Ours) | GCN | 0.58±0.009 | 0.4052±0.0221 | **1.29±0.07 (0.69, 0.60)** | **0.64±0.009** | 2.98±1.41 |

Table 10: Performance of counterfactual explanations on **Loan-Decision** and GCN.

| Method | Base GNN | Misclass. ↑ | Fidelity ↑ | $\Delta \mathbf{E}(\mathbf{E}^+, \mathbf{E}^-)$ ↓ | Plausibility ↑ | Time (sec) ↓ |
|---|---|---|---|---|---|---|
| **Explainers** | | | | | | |
| CF-GNNExplainer | GCN | 0.45±0.092 | 0.2520±0.0490 | 1.35±0.20 (0.00, 1.35) | 0.53±0.038 | 56.00±7.04 |
| INDUCE | GCN | 0.18±0.102 | 0.0873±0.0516 | 3.23±0.76 (1.21, 2.01) | 0.39±0.169 | 1.68±0.21 |
| C2Explainer | GCN | 0.00±0.000 | – | – | – | 39.88±1.23 |
| CF$^2$ | GCN | 0.17±0.005 | 0.0774±0.0038 | 3.58±0.08 (0.00, 3.58) | 0.43±0.012 | 31.76±0.57 |
| NSEG | GCN | 0.18±0.008 | 0.0891±0.0022 | 3.74±0.11 (0.00, 3.74) | 0.45±0.012 | 26.43±0.50 |
| GNNExplainer | GCN | 0.16±0.048 | 0.0438±0.0497 | 2.56±0.29 (0.00, 2.56) | 0.42±0.017 | 3.33±4.36 |
| PGExplainer | GCN | 0.10±0.008 | 0.0281±0.0105 | 2.80±0.34 (0.00, 2.80) | 0.21±0.024 | **0.32±0.04** |
| **Attack Models** | | | | | | |
| Nettack | GCN | 0.34±0.005 | 0.1685±0.0075 | 5.00±0.00 (3.01, 1.99) | 0.24±0.017 | 1.16±0.43 |
| GOttack | GCN | 0.35±0.005 | 0.1742±0.0053 | 5.00±0.00 (4.25, 0.75) | 0.15±0.005 | 0.52±0.16 |
| ATEX-CF (Ours) | GCN | **0.68±0.024** | **0.3658±0.0171** | **1.27±0.02 (0.38, 0.89)** | **0.67±0.026** | 20.33±0.58 |

outperforming all baselines by a large gap. These results confirm that our mask optimization generalizes beyond GCNs, remaining stable and effective across different architectures, including both attention-based and transformer-based GNNs.

Moreover, Tables 12 and 13 show that applying CF-GNNExplainer to attention-based models such as GAT and Graph Transformer often results in unstable mask optimization. This instability arises because, unlike GCN where the normalized adjacency enters linearly into the convolution allowing effective gradient flow from the loss to the mask, attention-based architectures compute edge attention coefficients via nonlinear transformations (LeakyReLU, softmax). Any mask applied to edge weights is absorbed and scaled by $\alpha_{ij}(1 - \alpha_{ij}) \ll 1$, leading to vanishing gradient signals and preventing the identification of meaningful counterfactual edges.

In CF-GNNExplainer, the adjacency mask $P$ is treated as a continuous parameter (after a sigmoid), which scales the edge weight or serves as an edge attribute. In attention-based models, these edge attributes enter the computation of attention logits $z_{ij}$:

$$z_{ij} = s_{ij} + b \cdot e_{ij}, \quad e_{ij} = \sigma(P_{ij}), \tag{10}$$

where $s_{ij}$ is a feature-derived score, $b$ is a scalar, and $\sigma$ is the sigmoid. The normalized attention coefficient is

$$\alpha_{ij} = \frac{\exp(z_{ij})}{\sum_{k \in \mathcal{N}(i)} \exp(z_{ik})}. \tag{11}$$

The gradient of the loss $L$ with respect to $P_{ij}$ is then

$$\frac{\partial L}{\partial P_{ij}} = \frac{\partial L}{\partial \alpha_{ij}} \cdot \underbrace{\frac{\partial \alpha_{ij}}{\partial z_{ij}}}_{\alpha_{ij}(1-\alpha_{ij})} \cdot \underbrace{\frac{\partial z_{ij}}{\partial e_{ij}}}_{b} \cdot \underbrace{\frac{\partial e_{ij}}{\partial P_{ij}}}_{\sigma'(P_{ij})}. \tag{12}$$

The critical term is the Jacobian of the softmax:

$$\frac{\partial \alpha_{ij}}{\partial z_{ij}} = \alpha_{ij}(1 - \alpha_{ij}).$$

When the degree of node $i$ is $N$ and neighbors are similar, $\alpha_{ij} \approx 1/N$, thus $\alpha_{ij}(1-\alpha_{ij}) \approx \mathcal{O}(1/N)$. This means that the mask gradient is strongly diluted by $1/N$, which is further multiplied by the

Table 11: Performance of counterfactual explanations on **ogbn-arxiv** and GCN.

| Method | Base GNN | Misclass. ↑ | Fidelity ↑ | ΔE(E⁺, E⁻) ↓ | Plausibility ↑ | Time (sec) ↓ |
|---|---|---|---|---|---|---|
| **Explainers** | | | | | | |
| CF-GNNExplainer | GCN | 0.45±0.033 | 0.0791±0.0072 | 1.56±0.06 (0.00, 1.56) | 0.66±0.005 | 7.16±1.71 |
| INDUCE | GCN | 0.28±0.066 | 0.0130±0.0015 | 2.17±0.24 (1.12, 1.05) | 0.41±0.009 | 0.15±0.02 |
| C2Explainer | GCN | 0.88±0.009 | 0.2552±0.0091 | 1.62±0.05 (0.64, 0.98) | 0.70±0.008 | 8.61±1.70 |
| CF² | GCN | 0.42±0.012 | 0.0700±0.0100 | 2.85±0.04 (0.00, 2.85) | 0.48±0.021 | 6.99±0.05 |
| NSEG | GCN | 0.64±0.017 | 0.1956±0.0058 | 3.53±0.10 (0.00, 3.53) | 0.61±0.008 | 3.38±0.16 |
| GNNExplainer | GCN | 0.33±0.014 | 0.0136±0.0040 | 2.22±0.07 (0.00, 2.22) | 0.63±0.008 | **0.10±0.01** |
| PGExplainer | GCN | 0.26±0.012 | 0.0206±0.0056 | 2.25±0.13 (0.00, 2.25) | 0.59±0.022 | **0.10±0.05** |
| **Attack Models** | | | | | | |
| Nettack | GCN | 0.86±0.009 | **0.3366±0.0059** | 5.00±0.00 (4.38, 0.62) | 0.58±0.029 | 2.14±0.69 |
| GOttack | GCN | 0.85±0.022 | 0.3251±0.0107 | 5.00±0.00 (5.00, 0.00) | 0.62±0.012 | 0.63±0.08 |
| ATEX-CF (Ours) | GCN | **0.90±0.012** | 0.3251±0.0023 | **1.20±0.05 (0.92, 0.28)** | **0.73±0.017** | 3.35±0.17 |

Table 12: Performance of counterfactual explanations on **Loan-Decision** and Graph Transformer.

| Method | Base GNN | Misclass. ↑ | Fidelity ↑ | ΔE(E⁺, E⁻) ↓ | Plausibility ↑ | Time (sec) ↓ |
|---|---|---|---|---|---|---|
| CF-GNNExplainer | Graph Trans. | – | – | – | – | – |
| INDUCE | Graph Trans. | 0.02±0.012 | 0.1264±0.0044 | 3.57±0.43 (2.10, 1.47) | 0.29±0.000 | 0.35±0.09 |
| C2Explainer | Graph Trans. | 0.06±0.009 | 0.1549±0.0165 | 2.95±0.16 (2.36, 0.59) | **0.53±0.033** | 8.02±0.29 |
| CF² | Graph Trans. | 0.05±0.005 | 0.1402±0.0052 | 3.61±0.07 (0.00, 3.61) | 0.46±0.021 | 7.03±0.07 |
| NSEG | Graph Trans. | 0.07±0.012 | 0.1701±0.0245 | 3.89±0.07 (0.00, 3.89) | 0.44±0.022 | 4.10±0.10 |
| GNNExplainer | Graph Trans. | 0.30±0.039 | 0.2452±0.0513 | 2.99±0.27 (0.00, 2.00) | 0.36±0.012 | 4.77±3.45 |
| PGExplainer | Graph Trans. | 0.39±0.007 | 0.3187±0.0141 | **1.66±0.27 (0.00, 1.66)** | 0.45±0.031 | **0.05±0.03** |
| Nettack | Graph Trans. | 0.32±0.004 | 0.2509±0.0101 | 5.00±0.00 (4.03, 0.97) | 0.22±0.016 | 1.03±0.48 |
| GOttack | Graph Trans. | 0.31±0.006 | 0.2420±0.0072 | 5.00±0.00 (4.81, 0.19) | 0.30±0.005 | 0.66±0.21 |
| ATEX-CF (Ours) | Graph Trans. | **0.44±0.035** | **0.3563±0.0120** | **1.66±0.01 (0.85, 0.81)** | 0.50±0.024 | 8.07±2.41 |

sigmoid derivative $\sigma'(P_{ij})$. As a result, the gradient magnitude quickly vanishes, especially for high-degree nodes. This explains why CF-GNNExplainer struggles to optimize adjacency masks in attention-based models.

In contrast, ATEX-CF uses a *signed, discrete mask* $M_{ij} \in \{-1, 0, +1\}$ combined with a straight-through estimator (STE) for backpropagation:

$$\tilde{A}_{ij} = \tilde{A}_{ij}^{\text{discrete}} + \left(M_{ij}^{\text{cont}} - \text{sg}(M_{ij}^{\text{cont}})\right), \quad \frac{\partial L}{\partial M_{ij}^{\text{cont}}} \approx \frac{\partial L}{\partial \tilde{A}_{ij}}, \tag{13}$$

where $\tilde{A}_{ij}$ is the perturbed adjacency entry for edge $(i, j)$, $\tilde{A}_{ij}^{\text{discrete}}$ is the discrete (binary) adjacency entry, $M_{ij}^{\text{cont}}$ is the continuous mask, $\text{sg}(\cdot)$ denotes the stop-gradient operator that blocks gradients, and $L$ is the model loss. This allows gradients to capture the *finite-difference effect* of adding or deleting an edge on the loss, without attenuation from softmax or nonlinearities. As a result, the signed mask optimization remains stable and effective across both GCNs and attention-based models, enabling reliable counterfactual explanations.

## A.8 A CASE STUDY ON COUNTERFACTUALS

To illustrate the refinement effect of pruning, Figure 7 compares the initial and final counterfactual explanations for a target node. Before the pruning, the explanation includes several edges of varying influence, some of which are unnecessary for achieving the prediction flip. The attack procedure discards these redundant edits by greedily testing their necessity. In this case, the final counterfactual achieves its goal by deleting the edge $(1978, 1306)$ and adding a single edge $(1978, 190)$. This minimal perturbation is sufficient to alter the model's decision.

## A.9 ABLATION STUDY

Table 15 shows the effect of removing each loss on Cora. Removing $\mathcal{L}_{dist}$ reduces misclassification slightly $(0.71 \to 0.70)$, leading to larger edit sets $(1.62 \to 1.66)$ and lower plausibility $(0.75 \to 0.71)$, indicating that edit minimality is compromised. Omitting the plausibility loss ($\mathcal{L}_{plau}$) yields the smallest edit size (1.57), but severely hurts misclassification, dropping the rate to 0.68, as edits no longer respect semantic structure. Removing both losses reduces misclassification and plausibility.

Table 13: Performance of counterfactual explanations on **Loan-Decision** and GAT.

| Method | Base GNN | Misclass. ↑ | Fidelity ↑ | $\Delta\mathbf{E}(\mathbf{E}^+, \mathbf{E}^-)$ ↓ | Plausibility ↑ | Time (sec) ↓ |
|---|---|---|---|---|---|---|
| CF-GNNExplainer | GAT | – | – | – | – | – |
| INDUCE | GAT | 0.18±0.115 | 0.0420±0.0280 | 2.64±0.18 (1.63, 1.01) | 0.38±0.061 | 0.24±0.01 |
| C2Explainer | GAT | 0.08±0.021 | 0.0108±0.0055 | 3.03±0.18 (2.46, 0.57) | 0.52±0.033 | 6.42±0.09 |
| CF$^2$ | GAT | 0.07±0.008 | 0.0126±0.0029 | 3.35±0.04 (0.00, 3.35) | 0.47±0.017 | 5.11±0.09 |
| NSEG | GAT | 0.12±0.008 | 0.0297±0.0014 | 3.68±0.09 (0.00, 3.68) | 0.45±0.016 | 3.13±0.07 |
| GNNExplainer | GAT | 0.01±0.005 | 0.0002±0.0315 | 3.00±0.00 (0.00, 3.00) | 0.34±0.015 | 3.40±1.67 |
| PGExplainer | GAT | 0.08±0.032 | 0.0057±0.0012 | 3.39±0.00 (0.00, 3.39) | 0.46±0.092 | **0.06±0.01** |
| Nettack | GAT | 0.41±0.006 | 0.0781±0.0078 | 5.00±0.12 (3.93, 1.07) | 0.20±0.014 | 1.01±0.51 |
| GOttack | GAT | 0.32±0.007 | 0.0689±0.0043 | 5.00±0.04 (4.80, 0.20) | 0.18±0.005 | 0.60±0.18 |
| ATEX-CF (Ours) | GAT | **0.47±0.021** | **0.0892±0.0193** | **1.58±0.03 (0.67, 0.91)** | **0.65±0.019** | 4.32±1.95 |

Table 14: Performance of counterfactual explanations on **Chameleon** and GCN.

| Method | Base GNN | Misclass. ↑ | Fidelity ↑ | $\Delta\mathbf{E}(\mathbf{E}^+, \mathbf{E}^-)$ ↓ | Plausibility ↑ | Time (sec) ↓ |
|---|---|---|---|---|---|---|
| **Explainers** | | | | | | |
| CF-GNNExplainer | GCN | 0.39±0.033 | 0.0858±0.0134 | 1.81±0.02 (0.00, 1.81) | **0.81±0.054** | 375.39±8.38 |
| INDUCE | GCN | 0.03±0.005 | 0.0096±0.0004 | 2.33±0.10 (1.50, 0.83) | 0.44±0.033 | 18.23±0.29 |
| C2Explainer | GCN | 0.09±0.016 | 0.0596±0.0026 | 2.61±0.07 (2.17, 0.44) | 0.76±0.008 | 85.69±1.53 |
| CF$^2$ | GCN | 0.21±0.025 | 0.0879±0.0091 | 3.51±0.12 (0.00, 3.51) | 0.62±0.024 | 73.31±2.40 |
| NSEG | GCN | 0.30±0.025 | 0.0870±0.0069 | 3.86±0.09 (0.00, 3.86) | 0.61±0.021 | 42.25±1.54 |
| GNNExplainer | GCN | 0.01±0.005 | 0.0046±0.0023 | 3.50±0.41 (0.00, 3.50) | 0.53±0.008 | 11.28±0.28 |
| PGExplainer | GCN | 0.01±0.005 | 0.0003±0.0002 | 2.83±2.09 (0.00, 2.83) | 0.45±0.328 | **1.79±0.10** |
| **Attack Models** | | | | | | |
| Nettack | GCN | 0.51±0.008 | 0.1806±0.0008 | 5.00±0.00 (4.73, 0.27) | 0.29±0.012 | 38.23±0.70 |
| GOttack | GCN | 0.52±0.029 | 0.1944±0.0010 | 5.00±0.00 (4.87, 0.13) | 0.43±0.012 | 9.26±0.13 |
| ATEX-CF (Ours) | GCN | **0.84±0.008** | **0.2595±0.0004** | **1.64±0.01 (1.16, 0.48)** | 0.76±0.005 | 134.73±4.15 |

These findings demonstrate that $\mathcal{L}_{dist}$ enforces concise edits, $\mathcal{L}_{plau}$ preserves semantic plausibility, and their combination in ATEX-CF achieves the best overall balance across all metrics.

**Ablation on Attack Candidate Selection.** We conducted ablations by replacing the GOttack-based candidate selection with two alternative strategies: (i) random sampling and (ii) Nettack-based edge proposals. These modifications isolate the impact of attack-informed candidate generation on counterfactual quality.

As summarized in Table 16, using GOttack candidates yields the highest misclassification rate and fidelity under the same budget, outperforming both random and Nettack-based baselines. Compared to random sampling, GOttack offers a $\approx 50\%$ increase in flip rate with higher plausibility and lower runtime. This demonstrates that attack-informed candidates meaningfully improve counterfactual effectiveness.

### A.10 SENSITIVITY ANALYSIS

We analyze the key hyperparameters using Cora. **Search depth** ($l$): Varying the number of hops for local structure surrounding the target node shows diminishing returns beyond local context. Figure 4 depicts that going from $l = 2$ to $l = 3$ yields only marginal improvements in fidelity, edits, and plausibility (e.g., +0.06 fidelity, -0.38 $\Delta\mathbf{E}$, +0.03 plausibility), while increasing computation time and dropping misclassification. This indicates that depth-2 captures sufficient structure for effective counterfactuals. **Hyperparameters** ($\alpha_{deg}, \alpha_{motif}$): We vary the weights of degree-anomaly and motif-anomaly terms in plausibility loss. Figure 5 demonstrates that ATEX-CF is robust across a range of values (e.g. $\alpha = 0.5$–$1.5$); misclassification and fidelity remain high. Very low $\alpha$ removes the corresponding regularizer and slightly degrades plausibility, while very high $\alpha$ yields negligible gains but more aggressive edits. In practice, moderate $\alpha$ maximizes fidelity and plausibility together.

### A.11 IMPACT AND BENEFIT OF THE PRUNING STRATEGY

We also evaluate the impact of our candidate-edge pruning on GCN with the Cora dataset. As shown in Figure 6, pruning yields more concise explanations by reducing redundant edits ($\Delta\mathbf{A} = 1.71 \rightarrow 1.62$), while maintaining nearly identical predictive accuracy (misclassification = 0.71)

Table 15: Ablation Study on **Cora** and GCN.

| Method | Base GNN | Misclass. ↑ | Fidelity ↑ | $\Delta$E($E^+$, $E^-$) ↓ | Plausibility ↑ | Time (sec) ↓ |
|---|---|---|---|---|---|---|
| w/o $\mathcal{L}_{dist}$ | GCN | 0.70 | 0.2360 | 1.66 (0.76, 0.90) | 0.71 | 7.42 |
| w/o $\mathcal{L}_{plau}$ | GCN | 0.68 | 0.2225 | **1.57 (0.72, 0.85)** | 0.71 | 6.82 |
| w/o $\mathcal{L}_{dist}$ and $\mathcal{L}_{plau}$ | GCN | 0.69 | **0.2469** | 1.60 (0.59, 1.01) | 0.68 | 6.22 |
| ATEX-CF (Ours) | GCN | **0.71** | 0.2336 | 1.62 (0.92, 0.70) | **0.75** | **3.80** |

Figure 4: Performance of counterfactual explanations vs. the number of GNN layers: The results demonstrate sensitivity w.r.t. the number of hops for the local structure surrounding the target node.

and plausibility (0.76 vs. 0.75). Runtime is significantly reduced (6.12s vs. 3.00s), confirming that pruning improves explanatory minimality and efficiency without sacrificing fidelity or plausibility.

We found optimizing the loss directly preferable to pruning for two reasons. First, the loss computation is efficient and integrates naturally into gradient-based optimization. Second, pruning after proposing a perturbation can be counterproductive in graph settings, where a single edge change can change the receptive fields of many nodes.

We illustrate the second point with an example. Suppose that the model proposes the sequence of edits: $+e_7$, $+e_8$, $-e_2$. If $+e_7$ is deemed implausible and later pruned, the remaining edits $+e_8$ and $-e_2$ may no longer have any effect, as their impact depended on $+e_7$ being applied. This leads to wasted computation and potentially invalid counterfactuals. In contrast, our loss-based optimization penalizes implausible edits during training. As a result, $+e_7$ would be discarded early, and the model would avoid proposing follow-up edits that rely on its presence. This allows us to generate counterfactuals more efficiently and with greater consistency.

In the first reason, we noted that plausibility loss is computationally cheap. This is because ATEX-CF restricts all computations to the $((l + 1)$-hop subgraph) of the target node (e.g., GCN's 3 layers + 1), the maximum degree $(d_{max})$ is naturally bounded by this local neighborhood.

Under this setting, the plausibility loss remains computationally lightweight. Even in worst-case large graphs such as ogbn-arxiv, the 4-hop neighborhood typically has $(d_{max} \approx 50, n_{sub} \approx 1000)$, amounting to $(1000 \times 50^2 = 2.5 \times 10^6)$ operations, still negligible compared to a single GNN forward pass $O(L \times |E|d + L \times n_{sub} \times d^2)$. Other datasets' numbers are given in Table 17.

Pruning is costly because it requires a GNN forward pass to detect which edges to perturb. Integrating the plausibility loss helps us avoid unnecessary, costly passes for candidates that would be pruned at the end.

### A.12 Proof and Evidence for the Hypotheses

Throughout this section, let $v$ be the target node under analysis. We use $s_G(v)$ to denote the logit of the target class for $v$, $f_G(v)$ for the predicted label of $v$, and $m_v$ for the margin between the logit of $v$'s true class and the highest competing class.

By definition, $CFEx(G)$ is an inclusion-minimal set of edge modifications (additions or deletions) such that applying them flips $f$'s prediction for node $v$. Minimal means that no proper subset of

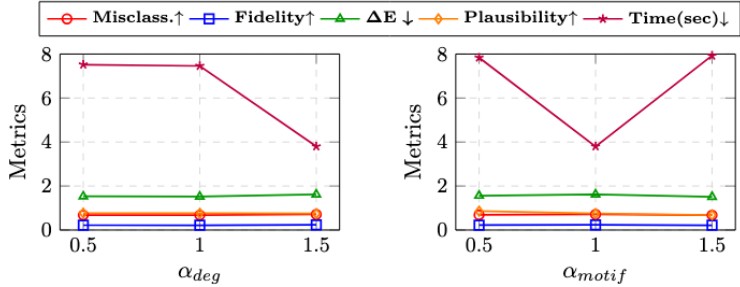

Figure 5: Sensitivity w.r.t. Hyperparameters $\alpha_{deg}$ and $\alpha_{motif}$.

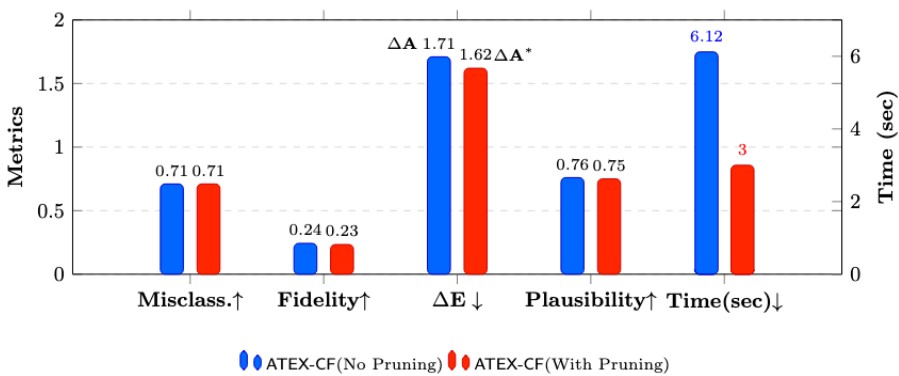

Figure 6: Effectiveness of Post-Hoc Pruning.

those modifications is sufficient to flip the prediction. Let us denote this modification set by $F :=$ $CFEx(G)$.

$$f_{G \oplus F}(v) \neq f_G(v),$$

but for any strict subset $F' \subsetneq F$,

$$f_{G \oplus F'}(v) = f_G(v).$$

We assume that the influence function of $f$ over edge sets is submodular, so the marginal effect of adding or removing an edge diminishes as more modifications are applied (influence functions on graphs are often modeled as submodular (Krause & Guestrin, 2007; Borgs et al., 2014)). This submodularity assumption implies that the minimal counterfactual explanation set $F$ is unique, which ensures that alignment between the attack-selected edges and the explanation subgraph is well defined, i.e., the top-$k$ edges chosen by the attack coincide with the uniquely defined set $F$ rather than an arbitrary minimal set.

When multiple such minimal sets exist, we fix a canonical choice by breaking ties, for example, by selecting the lexicographically smallest edge set. Intuitively, $F$ captures the single most crucial evidence subgraph in $G$ supporting the original prediction.

For any edge $e$ and set of edges $S$, define the conditioned marginal effect as $\Delta_e f(G \cup S; v) := f_{G \cup S \cup \{e\}}(v) - f_{G \cup S}(v)$.

### A.12.1 HYPOTHESIS H1: EDGE GRADIENT ATTACK ALIGNMENT

**Hypothesis 1** (Restated). *Let $G = (A, X)$ be an input graph and $f$ a pre-trained GNN classifier. For a target node $v$, let $\Delta G(E^+)$ denote the set of added edges in an evasion attack that flips the prediction of $f$, and let $CFEx(G)$ denote the counterfactual explanation graph of the graph $G$. Then, the graph similarity between $\Delta G(E^+)$ and $CFEx(G)$:*

$$Sim(\Delta G(E^+), CFEx(G)) \approx c, \quad 0 << c < 1,$$

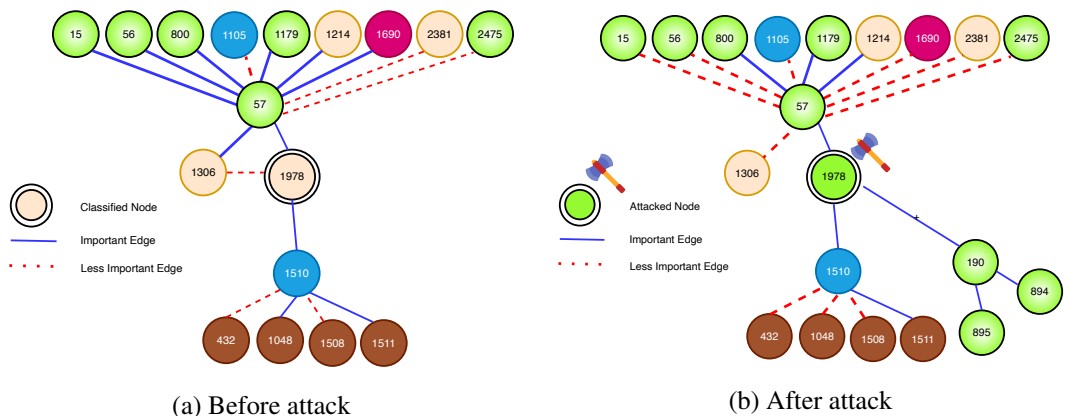

(a) Before attack          (b) After attack

Figure 7: Visual comparison of the changed graph when the edge 1978 is classified (panel a) and its counterfactual edges (panel b). Node colors indicate class membership. The initial classification includes both important and less important edges. The counterfactual deletes edge (1978, 1306) and adds edge (1978, 190) to successfully flip the prediction for 1978.

Table 16: Effect of attack candidate generation strategy on counterfactual explanations (Cora, GCN, $\kappa = 5$).

| Method | Base GNN | Misclass. | Fidelity | $\Delta E$ ($E^+$, $E^-$) | Plausibility | Time (sec) |
|---|---|---|---|---|---|---|
| ATEX-CF + Random Sampling | GCN | 0.46 | 0.17 | 1.71 (0.74, 0.97) | 0.67 | 7.44 |
| ATEX-CF + Nettack | GCN | 0.67 | 0.22 | 1.62 (0.75, 0.87) | 0.72 | 8.97 |
| ATEX-CF + GOttack (ours) | GCN | 0.69 | 0.22 | 1.66 (0.76, 0.90) | 0.72 | 4.71 |

*where $Sim(\cdot, \cdot)$ denotes a graph similarity measure by graph edit distance, maximum common subgraph, and graph embedding vectors, and $c$ is a positive score, indicating non-trivial overlap between the attack edges and the explanation graph.*

**Proof Sketch:** Edges with the largest gradient influence on the target node's logit margin are the most potent for adversarial attacks. Formally, for a target node $v$ with margin $m_v(A)$ and edge gradients $g_e = \partial m_v/\partial A_e$, suppose $e_1$ and $e_2$ are two candidate edges (with $e_1$ either currently present or absent depending on the attack type, and similarly for $e_2$). If $|g_{e_1}| > |g_{e_2}|$, then flipping $e_1$ (adding it if $g_{e_1} < 0$ or removing it if $g_{e_1} > 0$) yields a larger drop in $m_v$ than flipping $e_2$. In particular, a Projected Gradient Descent (PGD) attack will primarily select edges from among those with the highest $|g_e|$, aligning adversarial modifications with the gradient-based explanation subgraph. Intuitively, the gradient $g_e$ indicates how sensitively the margin $m_v$ changes with respect to edge $e$. A large-magnitude gradient $|g_e|$ means that a small change in $A_e$ has a big effect on $m_v$. In a 2-layer GCN with ReLU, the model is piecewise linear, so locally $m_v$ changes approximately linearly with $A_e$. Thus, the edge with the largest $|g_e|$ produces the steepest change in $m_v$ when perturbed. A PGD adversarial attack, which follows the gradient of the loss (or negative margin), will therefore choose the edge with the most negative gradient (for additions) or the most positive gradient (for deletions) to maximally decrease the margin. In essence, explanation methods pick out these high-$|g_e|$ edges as important, and the attacker targets the very same edges to flip the prediction.

*Proof.* Consider the target node $v$ with true class $y_v$ and margin $m_v(A) = z_{y_v}(A, v) - \max_{c \neq y_v} z_c(A, v)$. Let $g_e = \frac{\partial m_v}{\partial A_e}$ be the gradient influence of edge $e$ on the margin. We analyze edge addition, and show that larger $|g_e|$ implies a greater reduction in margin when $e$ is perturbed:

**Edge addition (E+ attack).** Suppose $e = (i, j)$ is a non-existent edge ($A_e = 0$). If $g_e < 0$, then $e$ is a detrimental or counterfactual edge for the current prediction: increasing $A_e$ (adding this edge) will lower the margin $m_v$. In a small continuous relaxation of $A_e$, $m_v$ would decrease by about $|g_e| \cdot \Delta A_e$. For the actual discrete addition ($A_e : 0 \rightarrow 1$), the change $m_v(A_{+e}) - m_v(A)$ will be approximately $g_e$ (since $g_e$ is negative, this is a drop in margin). Because our GCN is piecewise linear (ReLU activation), adding $e$ causes a margin change on the order of $g_e$. If $|g_{e_1}| > |g_{e_2}|$ for

Table 17: Graph statistics and localized density parameters across datasets.

| Dataset | Global Avg. Degree | Global Max Degree | Typical 4-hop Local Max | Reasonable $d_{\max}$ | Typical $n_{\text{sub}}$ |
|---------|-------------------|-------------------|------------------------|----------------------|-------------------------|
| BA-SHAPES | $\sim$4 | $<$10 | 4–6 | 5 | 50–100 |
| Cora | $\sim$3.7 | $\sim$40 | 12–16 | 15 | 100–300 |
| Ogbn-arxiv | $\sim$13 | $\sim$1800 | 40–60 | 50 | 500–1000 |

Table 18: **Attacks and Counterfactuals.** The structural similarity between evasion attack edges $\Delta\mathbf{G}$ (mainly additions $\Delta\mathbf{E}^+$ from GOttack) and instance-level factual explanations $Ex(G')$ from GNNExplainer on post-attack graph $G'$. 280 target nodes are correctly classified in the original graph $G$. Budget = 5. GCN (2-layer), **Cora** dataset.

| Metric | All (280) | Attack Success (225) | Attack Fail (55) |
|--------|-----------|---------------------|------------------|
| GED↓ | 0.38 | 0.37 | 0.41 |
| MCS↑ | 0.31 | 0.33 | 0.24 |
| GEV↑ | 0.72 | 0.80 | 0.39 |

two absent edges with negative gradients, adding $e_1$ produces a larger margin drop than adding $e_2$. Thus, an adversary performing PGD will add the edge with the most negative gradient first, which is precisely the top edge identified by a counterfactual explanation method.

The attacker's choice of edge corresponds to the edge with the largest $|g_e|$ that reduces the margin (negative $g_e$ for addition). By repeating this argument iteratively (considering the next most influential edge after the first, and so on), one can see that an attack adding/removing $k$ edges will choose the $k$ edges with highest gradient magnitudes that contribute to lowering $m_v$. Therefore, the set of edges targeted by the PGD attack aligns with the gradient-based counterfactual explanation subgraph (which consists of edges with the largest $|g_e|$). This establishes that ranking edges by $|g_e|$ is equivalent to ranking them by adversarial effectiveness, proving the hypothesis. $\square$

**Empirical Evidence for Hypothesis 1**

The results in Table 18 support Hypothesis 1, which posits a high structural overlap between the attacker's perturbation $\Delta G$ and the counterfactual explanation $CFEx(G)$ produced by pre-attack explanation methods. Notice that here we consider the instance-level factual explanations $Ex(G')$ from GNNExplainer (Ying et al., 2019) on the post-attack graph $G'$ as a proxy for the counterfactual explanation $CFEx(G)$ produced by pre-attack explanation methods. This is because the state-of-the-art counterfactual explainers generally do not support edge addition.

In both correctly and incorrectly predicted instances, the Graph Edit Distance (GED) remains moderate ($\approx 0.38$), and the Maximum Common Subgraph (MCS) similarity is non-negligible, particularly for successful attacks. Notably, Graph Embedding Vector (GEV) similarity reaches 0.88 for misclassified nodes and 0.80 for successful attacks on correctly predicted nodes, indicating substantial alignment in the embedded subgraph structure. In other words, Table 18 shows that similarity between attack perturbations $\Delta G$ and counterfactual explanations $CFEx(G)$ depends strongly on attack outcome. For successful attacks, distances such as GED are lower (lower is better) and similarities such as MCS and GEV are higher (higher is better), while for failed attacks, the opposite holds. In other words, when the attack succeeds, the perturbations align closely with counterfactual explanations, whereas in failed cases the overlap weakens. This pattern offers evidence that effective adversarial edits not only cause misclassification but also resemble the explanatory structures that counterfactual methods would identify.

### A.12.2 PROPOSITIONS ON COUNTERFACTUAL COMPLETENESS VIA ATTACK-INFORMED ADDITIONS

In principle, for the completeness of our hypothesis, one would like to prove that edge additions "always" yield a successful counterfactual attack, which would strengthen our claim that unifying

attacks and counterfactuals is universally beneficial, even when counterfactuals alone fail. Unfortunately, this cannot be guaranteed, since the data may lack any node whose connection to the target would flip its label. Instead, we establish a next-best guarantee: When sufficiently informative opposite-class nodes exist, additions can flip the label while deletions cannot. State-of-the-art counterfactual explanations may overlook such opportunities, but attack algorithms are designed to exploit them.

Let $f$ be a GNN classifier and let $v \in V$ be a target node with $f_G(v) = y$. Throughout, $s_G(v)$ denotes a real valued class $y$ score for $v$, $f_G(v)$ denotes the predicted label, and $m_v$ denotes the margin for class $y$ at $v$. $w_{vu}$ is the weight assigned by the model to the contribution of neighbor $u$ when aggregating into the score of node $v$. Fix a one versus rest view for class $y$ and use the decision rule $f_G(v) = y$ if and only if $s_G(v) > 0$. Assume an additive, degree-independent neighborhood model

$$s_G(v) = bias_v + \sum_{u \in \mathcal{N}(v)} w_{vu} \, r_u,$$

with $w_{vu} \geq 0$, where $r_u$ is the contribution aligned with class $y$. This additive influence model abstracts away normalization and attention redistribution, but shows the monotonic nature of homophilic neighborhoods. While GNNs are more complex, we observe empirically that their behavior is consistent with the model's prediction: deletion of a few homophilic neighbors rarely flips predictions, whereas a small number of targeted additions frequently does (as evidenced in the addition attacks of Gottack (Alom et al., 2025) and Nettack (Zügner et al., 2018)).

We assume homophily in the immediate neighborhood so that $f_G(u) = y$ for all $u \in \mathcal{N}(v)$, hence $r_u \geq 0$ for all incident neighbors. No term in $s_G(v)$ is rescaled by $|\mathcal{N}(v)|$.

In this setting, deletion and addition have asymmetric effects. Deleting any number of incident edges can only remove nonnegative summands, while adding edges to informative opposite class nodes can introduce negative summands. The next two propositions formalize this.

**Proposition A.1** (Deletion Infeasibility). *Let $G' = G \setminus S$ for some strict subset $S \subsetneq (v, u) : u \in \mathcal{N}(v)$. If*

$$bias_v + \min_{u \in \mathcal{N}(v)} w_{vu} \, r_u \, > \, 0,$$

*where $bias_v$ is a bias term for node $v$'s own features, $r_u$ is the contribution from neighbor $u$'s features, aligned with class $y$, and $w_{vu} \geq 0$ is the scalar weight that measures how strongly neighbor $u$ influences $v$'s score. Then $f_{G'}(v) = y$. In words, as long as at least one incident neighbor remains, the score stays positive, and the label does not change.*

*Argument.* The smallest possible post-deletion score over all strict subsets occurs when only the least contributing neighbor of $v$ remains. This score equals $b_v + \min_u w_{vu} r_u$, which is positive by assumption, hence $f_{G'}(v) = y$.

**Proposition A.2** (Addition Sufficiency). *Suppose there exists a set of candidate nodes $C$ with $f_G(u) \neq y$ such that for each $u \in C$, adding the edge $(v, u)$ decreases the score by at least a fixed amount $\gamma > 0$:*

$$s_{G \cup \{(v,u)\}}(v) \, \leq \, s_G(v) - \gamma.$$

*Let $m_v = s_G(v) > 0$. Then there exists a set $E^+ \subseteq (v, u) : u \in C$ with*

$$|E^+| \, \leq \, \lceil m_v / \gamma \rceil$$

*such that $f_{G \cup E^+}(v) \neq y$. Thus, a small number of informative additions flips the prediction.*

*Argument.* Each addition reduces the score by at least $\gamma$. After $k = \lceil m_v / \gamma \rceil$ additions, the score is nonpositive, which changes the predicted label.

Table 19: Failure rate of deletion-based counterfactual explanations for correctly predicted target nodes (**Cora**, 2-layer GCN).

| Method | Total Nodes | Has CF Explanation | No CF Explanation |
|---|---|---|---|
| CF-GNNExplainer | 280 | 54 | 226 |

Table 20: Failure rate of deletion-based counterfactual explanations for incorrectly predicted target nodes (**Cora**, 2-layer GCN).

| Method | Total Nodes | Has CF Explanation | No CF Explanation |
|---|---|---|---|
| CF-GNNExplainer | 220 | 76 | 144 |

**Corollary A.3** (Budgeted reachability and strict advantage of additions)**.** *Let* $\mathcal{R}_{\mathrm{del}}(k) = \{G \setminus S :$ $S \subseteq \{(v, u) : u \in \mathcal{N}(v)\}, |S| \leq k\}$ *and* $\mathcal{R}_{\mathrm{add}}(k) = \{G \cup E^+ : E^+ \subseteq \{(v, u) : u \in C\}, |E^+| \leq$ $k\}$*. Under the assumptions above, if*

$$bias_v + \min_{u \in \mathcal{N}(v)} w_{vu} r_u > 0,$$

*then for every* $k < |\mathcal{N}(v)|$ *there is no graph in* $\mathcal{R}_{\mathrm{del}}(k)$ *that flips* $v$*'s label. If, in addition, there exists* $\gamma > 0$ *such that each* $(v, u)$ *with* $u \in C$ *decreases* $s_G(v)$ *by at least* $\gamma$*, then with* $k_+ = \lceil m_v/\gamma \rceil$ *there exists* $G^+ \in \mathcal{R}_{\mathrm{add}}(k_+)$ *that flips* $v$*'s label. Consequently, whenever* $k_+ < |\mathcal{N}(v)|$*, the set of counterfactuals reachable by at most* $k_+$ *additions is nonempty while the set reachable by at most* $k_+$ *deletions is empty, hence additions strictly dominate deletions under equal edit budgets.*

*Argument.* The deletion claim follows from the deletion infeasibility proposition. The addition claim follows from the addition sufficiency proposition with $k_+ = \lceil m_v/\gamma \rceil$. If $k_+ < |\mathcal{N}(v)|$, then $\mathcal{R}_{\mathrm{add}}(k_+)$ contains a prediction flipping graph while $\mathcal{R}_{\mathrm{del}}(k_+)$ does not.

**Corollary A.4** (Edit cost and latent stability)**.** *Let* $d_{\mathrm{edit}}$ *be the edge edit distance. Any witnessing addition set* $E^+$ *has* $d_{\mathrm{edit}}(G, G \cup E^+) = |E^+| \leq \lceil m_v/\gamma \rceil$*. If a node-level embedding map* $\psi(v; G)$ *is* $L$*-Lipschitz with respect to incident edge edits at* $v$*, then*

$$\|\psi(v; G) - \psi(v; G \cup E^+)\|_2 \leq L |E^+| \leq L \lceil m_v/\gamma \rceil.$$

*Thus the latent perturbation can be bounded linearly by the required number of additions.*

*Remark.* The strict advantage condition $k_+ < |\mathcal{N}(v)|$ is testable from estimates of $m_v$ and per edge gains. If $k_+ \geq |\mathcal{N}(v)|$, the theory is agnostic about dominance, but the separation holds whenever the margin-to-gain ratio is small relative to the neighborhood size.

**Empirical Evidence for the Counterfactual Completeness**

Tables 19 and 20 show how often CF-GNNExplainer (Lucic et al., 2022) fails to generate deletion-only counterfactual explanations under two conditions.

In Table 19 (correctly predicted target nodes), out of 280 test nodes, only the "HAS CF EXPLA-NATION" column reports 54 nodes ($\approx 19\%$) for which a deletion-based counterfactual exists; the remaining 226 nodes ($\approx 81\%$) are in the "NO CF EXPLANATION" column. Similarly, in Table 20 (misclassified nodes), 76 out of 220 nodes ($\approx 35\%$) have a deletion-only counterfactual, while 144 nodes ($\approx 65\%$) do not.

These high failure rates support our theoretical propositions and corollaries: namely, that there are many nodes for which deletion-based counterfactuals are infeasible. These empirical gaps justify the necessity of incorporating attack-informed edge additions to recover explanations for those nodes.

### A.13 MORE RESULTS ON ASYMMETRIC COSTS OF EDGE PERTURBATIONS

For completeness, we report additional results on asymmetric perturbation costs under smaller budgets ($\kappa = 5$ and $\kappa = 10$). As shown in Tables 21 and 22, the qualitative behavior observed in

Section 5.3 remains consistent across budgets. Increasing the relative cost of edge additions systematically shifts the optimizer toward deletion-heavy solutions, reducing the number of added edges and eliminating them entirely when $C$ exceeds perturbation budget $\kappa$. This induces predictable effects on counterfactual quality: both misclassification success and fidelity decline as additions are discouraged, whereas plausibility remains comparatively stable until additions are nearly disallowed. These results confirm that cost asymmetry provides a reliable control mechanism for shaping the perturbation profile, while also highlighting that excessively large $C$ can over-constrain the search space and impair explanation effectiveness.

Table 21: Counterfactual performance under asymmetric addition cost $C$ with $\kappa = 5$.

| Addition Cost | Deletion Cost | Misclass. | Fidelity | $\Delta E$ (E$^+$, E$^-$) | Plausibility | Time (sec) |
|---|---|---|---|---|---|---|
| 0.5 | 1.0 | 0.69 | 0.23 | 1.69(0.78,0.91) | 0.72 | 10.5 |
| 0.8 | 1.0 | 0.69 | 0.22 | 1.66(0.77,0.89) | 0.72 | 10.4 |
| 1.0 | 1.0 | 0.69 | 0.22 | 1.66(0.76,0.90) | 0.72 | 10.2 |
| 2.0 | 1.0 | 0.67 | 0.22 | 1.59(0.65,0.94) | 0.71 | 10.5 |
| 2.5 | 1.0 | 0.64 | 0.20 | 1.49(0.62,0.87) | 0.71 | 10.7 |
| 5.0 | 1.0 | 0.53 | 0.15 | 1.35(0.49,0.86) | 0.68 | 11.2 |
| 6.0 | 1.0 | 0.43 | 0.10 | 1.59(0.00,1.59) | 0.61 | 11.3 |

Table 22: Counterfactual performance under asymmetric addition cost $C$ with $\kappa = 10$.

| Addition Cost | Deletion Cost | Misclass. | Fidelity | $\Delta E$ (E$^+$, E$^-$) | Plausibility | Time (sec) |
|---|---|---|---|---|---|---|
| 0.5 | 1.0 | 0.70 | 0.23 | 1.78(0.78,1.00) | 0.72 | 5.9 |
| 0.8 | 1.0 | 0.70 | 0.23 | 1.78(0.77,1.01) | 0.71 | 6.0 |
| 1.0 | 1.0 | 0.70 | 0.23 | 1.78(0.77,1.02) | 0.71 | 7.4 |
| 3.0 | 1.0 | 0.70 | 0.23 | 1.77(0.66,1.11) | 0.69 | 7.7 |
| 5.0 | 1.0 | 0.68 | 0.22 | 1.71(0.61,1.10) | 0.69 | 7.8 |
| 7.0 | 1.0 | 0.68 | 0.22 | 1.70(0.59,1.10) | 0.69 | 7.8 |
| 10 | 1.0 | 0.54 | 0.15 | 1.42(0.49,0.93) | 0.68 | 8.2 |
| 11 | 1.0 | 0.42 | 0.10 | 1.78(0.00,1.78) | 0.62 | 8.5 |

## A.14 ACKNOWLEDGMENT

YZ and AK acknowledge support from the Novo Nordisk Foundation grant (NNF 22OC0072415). SBY acknowledges support from the National Science and Foundation of China (62402082) and Scientific and Technological Research Program of Chongqing Municipal Education Commission (KJQN202400637).