# OpenReview forum: "ATEX-CF: Attack-Informed Counterfactual Explanations for Graph Neural Networks"
_ICLR.cc/2026/Conference — ICLR 2026 Poster_

### Official Review · Reviewer_M2c7 · 2025-10-28

**Soundness:** 3
**Presentation:** 3
**Contribution:** 3
**Rating:** 6
**Confidence:** 3

**Summary:**

The paper proposes the ATEX-CF framework, which applies adversarial attack techniques to help with the generation of counterfactual explanations, which explain what factors must differ in order to produce a different result. This is within the context of graph neural networks, where the counterfactual explanation addresses the least amount of changes to the graph needed to alter the model’s prediction on a node in the graph. Adversarial attacks have overlap with counterfactual explanations as these attacks focus on undermining GNNs by making deliberate minor changes to a graph to change a node’s predicted class. ATEX-CF specifically incorporates the technique of adding edges to a graph that is used in adversarial attacks, in addition to the edge deletion techniques used in counterfactual explanations. By incorporating edge additions, ATEX-CF can also capture previously missing relations between nodes, by adding in the edge that denotes the possible/probable relationship. The paper additionally is the first to make the connection between counterfactual explanations and adversarial attacks, and leveraging the two together to perform a task.

**Strengths:**

The novelty or originality of the paper is a strength. This work is the first to make the connection between the techniques and goals of adversarial attacks and counterfactual explanations. Both involve either making or examining the minimal changes that can be made to a graph to change the outcome of a prediction made on that graph or its nodes. The paper leverages and highlights the similarity of the two methods in terms of objective to alter predictions and their approach of minimal alterations to the graph and consolidates them in order to improve on counterfactual explanations giving rise to their ATEX-CF framework. The usage of node deletion from traditional factual explanation generation and node addition from adversarial attacks enables the graph to capture previously missing relationships within the graph, by providing edges between nodes that have a plausible relation, and allows counterfactual explanations to overcome some limitations it previously faced with only node deletions such as overlooking the possibility of edges already missing within the graph. In addition to this, the paper incorporates sparsity and plausibility constraints into the framework in order to effectively add nodes into the graph with ATEX-CF. This paper can be significant in the way its method far outperformed other baselines and initiates a conversation of looking at counterfactual explanations with adversarial attacks and bridges a gap between the two topics which have evolved in isolation.

The quality of the paper does not lack either, it is a good quality paper, providing mathematical/theoretical backing for the proposed method ATEX-CF which is sound and logical and clearly sets up the problem definition and notation in section 2 so that the reader can follow along for the subsequent sections. The paper presents a quite comprehensive appendix to follow up on derivations of propositions in the paper and theory is formatted so that it is easy to read and follow. The paper is nicely structured such that the sections slowly build on and introduce the components involved in the ATEX-CF framework and the preliminary concepts and intuition for the framework are ordered so that the reader understands how the method is being developed. Also the inclusion of figures depicting the overall framework or examples of the interactions between nodes and edges help the reader gain intuition on the problem and solution being proposed by the paper. The framework employs preexisting techniques to tackle the defined problem such as GOttack which helps to generate candidate edges that would be used in adversarial attacks. Additionally, the authors make the source code involved available for others to reproduce the experiment and results. The experiments are performed on both synthetic and real world datasets demonstrating the ability of ATEX-CF to perform in real world situations and the usage of known benchmarking datasets/methods helps contextualize the performance of ATEX-CF. The experiment results support the notion of the proposed method’s effectiveness as it is able to outperform the baselines with the best (lowest) overall rank and wins 18/25 of the datasets, while other baselines don’t come close to this performance. Additionally, an ablation study is done to examine the effects of toggling different loss components in ATEX-CF and help outline what component influences the performance of the framework in a certain manner.

**Weaknesses:**

A minor weakness would be while the paper does mention the limitations to this study being the assumption of addition and deletion of edges in the graph always being possible when really, it may not be possible in some real world graphs, which would be the end goal of the framework to perform on, the limitation is only disclosed in the appendix. The paper would benefit from disclosing this information in the actual paper and providing some discussion on the limitation would inform the readers and allow them to understand the state of ATEX-CF. Although it is good to gear a future step towards eliminating this limitation which is mentioned in the section of the limitation in the appendix.

**Questions:**

What is the future with this framework or research, what are some future directions to take this or what further implications can be gleaned from this work?

---

> ### Author Response · Authors · 2025-11-21
> **Reviewer M2c7: Q1**
>
> We thank the reviewer for taking the time to assess our paper and for the constructive suggestions. It's rewarding to see the recognition of key aspects of our work, e.g.,  the novelty, the article’s quality, and the extent of our experiments. In the following, we will reply to the reviewers' questions/comments and have revised our manuscript accordingly, with changes highlighted in green.
>
> > Q1: “What is the future with this framework or research, what are some future directions to take this or what further implications can be gleaned from this work?”
>
> **Response**: Thank you for the question. Future directions include extending ATEX-CF to dynamic graphs, incorporating node feature perturbations, and applying the framework to real-world domains like healthcare and finance. We outline these in Section 6.

---

> ### Author Response · Authors · 2025-11-21
> **Reviewer M2c7: W1**
>
> > W1: “A minor weakness would be while the paper does mention the limitations to this study being the assumption of addition and deletion of edges in the graph always being possible when really, it may not be possible in some real world graphs, which would be the end goal of the framework to perform on, the limitation is only disclosed in the appendix. The paper would benefit from disclosing this information in the actual paper and providing some discussion on the limitation would inform the readers and allow them to understand the state of ATEX-CF. Although it is good to gear a future step towards eliminating this limitation which is mentioned in the section of the limitation in the appendix.”
>
> **Response**: Thanks for the suggestion. You raise an interesting point that we have not seen addressed in the literature yet; what happens if the budget is real valued with tailored costs for each perturbation type? For the perturbation budget constraint ||\delta A||0 =||\Delta E+|| + ||\Delta E-||≤κ, we changed the budget constraint to C*||\Delta E+|| + ||\Delta E-||≤κ by defining a given asymmetric cost parameter C. The results are shown in three tables below, where we adjust the cost of addition w.r.t. deletion in the range (0.5, 21.0) for three budgets (κ=5, 10, 20) in two tables below and the Table 4 of the PDF, respectively. If both addition and deletion are equally costly (e.g., row 3 of 1.0 and 1.0), the costs are symmetric. As the addition cost increases towards the budget, the sparsity objective would favor deletions.
>
> Table Counterfactual explanations of asymmetric cost C on Cora and GCN under κ=5
> | Addition Cost | Deletion Cost | Misclass. | Fidelity | ∆E(E+,E-) | Plausibility | Time(sec) |
> |-------------------|--------------------|--------------|-----------|------------|----------------|--------|
> | 0.5 | 1.0 | 0.69  | 0.23    | 1.69(0.78,0.91)  | 0.72  | 10.5 |
> | 0.8 | 1.0 | 0.69  | 0.22    | 1.66(0.77,0.89)  | 0.72  | 10.4 |
> | 1.0 | 1.0 | 0.69  | 0.22    | 1.66(0.76,0.90)  | 0.72  | 10.2 |
> | 2.0 | 1.0 | 0.67  | 0.22    | 1.59(0.65,0.94)  | 0.71  | 10.5 |
> | 2.5 | 1.0 | 0.64  | 0.20    | 1.49(0.62,0.87)  | 0.71  | 10.7 |
> | 5.0 | 1.0 | 0.53  | 0.15    | 1.35(0.49,0.86)  | 0.68  | 11.2 |
> | 6.0 | 1.0 | 0.43  | 0.10    | 1.59(0.00,1.59)  | 0.61  | 11.3 |
>
>
> Table Counterfactual explanations of asymmetric cost C on Cora and GCN under κ=10
> | Addition Cost | Deletion Cost | Misclass. | Fidelity | ∆E(E+,E-) | Plausibility | Time(sec) |
> |-------------------|--------------------|--------------|-----------|------------|----------------|--------|
> | 0.5 | 1.0 | 0.70  | 0.23    | 1.78(0.78,1.00)  | 0.72  | 5.9 |
> | 0.8 | 1.0 | 0.70  | 0.23    | 1.78(0.77,1.01)  | 0.71  | 6.0 |
> | 1.0 | 1.0 | 0.70  | 0.23    | 1.78(0.77,1.02)  | 0.71  | 7.4 |
> | 3.0 | 1.0 | 0.70  | 0.23    | 1.77(0.66,1.11)  | 0.69  | 7.7 |
> | 5.0 | 1.0 | 0.68  | 0.22    | 1.71(0.61,1.10)  | 0.69  | 7.8 |
> | 7.0 | 1.0 | 0.68  | 0.22    | 1.70(0.59,1.10)  | 0.69  | 7.8 |
> | 10  | 1.0 | 0.54  | 0.15    | 1.42(0.49,0.93)  | 0.68  | 8.2 |
> | 11  | 1.0 | 0.42  | 0.10    | 1.78(0.00,1.78)  | 0.62  | 8.5 |
>
> Table 4 Counterfactual explanations of asymmetric cost C on Cora and GCN under κ=20
> | Addition Cost | Deletion Cost | Misclass. | Fidelity | ∆E(E+,E-) | Plausibility | Time(sec) |
> |-------------------|--------------------|--------------|-----------|------------|----------------|--------|
> | 0.5 | 1.0 | 0.70  | 0.23    | 1.78(0.78,1.00)  | 0.72  | 6.1 |
> | 0.8 | 1.0 | 0.70  | 0.23    | 1.78(0.77,1.01)  | 0.71  | 6.1 |
> | 1.0 | 1.0 | 0.70  | 0.23    | 1.78(0.77,1.01)  | 0.71  | 10.3 |
> | 3.0 | 1.0 | 0.70  | 0.23    | 1.82(0.66,1.16)  | 0.69  | 10.7 |
> | 5.0 | 1.0 | 0.70  | 0.23    | 1.82(0.65,1.17)  | 0.69  | 10.9 |
> | 10  | 1.0 | 0.69  | 0.23    | 1.80(0.61,1.19)  | 0.69  | 11.0 |
> | 15  | 1.0 | 0.69  | 0.23    | 1.76(0.60,1.16)  | 0.69  | 11.0 |
> | 20  | 1.0 | 0.54  | 0.15    | 1.42(0.49,0.93)  | 0.68  | 11.5 |
> | 21  | 1.0 | 0.42  | 0.10    | 1.78(0.00,1.78)  | 0.62  | 11.7 |
>
> **Analysis of Asymmetric Weight Effects**: These results show that increasing the asymmetric addition cost C gradually shifts the perturbations from mixed edge additions/deletions toward deletion-dominant or deletion-only solutions. As C grows, the model relies less on expensive additions, leading to decreased misclassification ability (i.e., the ability to successfully flipping the prediction) and decreased fidelity, as well as slightly lower plausibility. This confirms that asymmetric weighting effectively controls the balance between addition and deletion operations, whereas overly large C values (e.g., C≥5) restrict additions (delete-only strategy) and degrade counterfactual quality.
>
> This discussion is added as a new Section 5.3 in the PDF with Table 4 for κ=20.
>
> Please note that an earlier review requested this as well, so we have colored the text red in the PDF with the review order in mind.

---

> ### Author Response · Authors · 2025-11-24
> **Additional responses**
>
> Thank you once again for your thoughtful review and constructive feedback.
>
> We have uploaded the latest version of our manuscript to OpenReview, where we have carefully addressed all of your previous concerns. In addition to the responses provided above, the revised submission includes the following updates:
>
> 1. Clarification of asymmetric cost results: We have expanded our discussion by adding Appendix A.13, which provides a more detailed explanation of the observed outcomes. (W1)
>
> We hope these revisions satisfactorily address your remaining questions and further strengthen your support for our work.

---

### Official Review · Reviewer_GcK9 · 2025-10-31

**Soundness:** 2
**Presentation:** 1
**Contribution:** 2
**Rating:** 4
**Confidence:** 4

**Summary:**

The authors propose ATEX-CF, a counterfactual explainer for GNNs that leverages adversarial insights by allowing both edge additions (common in attacks) and deletions (common in counterfactuals). This increases flip probability under a small perturbation budget and aids interpretability: deletions assess the impact of existing connections, additions probe plausible new ones. The objective balances flip, sparsity, and plausibility, with plausibility penalizing degree and local motif jumps. Experiments span BA-SHAPES, TREE-CYCLES, Loan-Decision, Cora, and ogbn-arxiv.

**Strengths:**

- Originality: The paper connects adversarial edge additions to counterfactual generation, which usually uses edge deletions instead.
- Solid optimization mechanics: The paper involves signed mask with ternary forward discretization, top-κ budget, and minimality-aware pruning that reduces edits/runtime while preserving flips/plausibility.
- Reproducibility and robustness checks: Code and configs released with means/SDs across seeds; sensitivity and pruning analyses quantify stability and efficiency.

**Weaknesses:**

- Theory: “Hypothesis 1” is asserted as “proved” in the appendix but functions more as an assumption tied to empirical overlap; this weakens guarantees.
- Plausibility metric is coarse: degree/clustering penalties may not capture domain constraints like temporal or type compatibility; evaluation reuses the same surrogate.
- Dependence on attack generator: Candidate additions come from GOttack; robustness to this choice and ablations vs. simpler heuristics are unclear.
- Scope: Only node classification; no feature-perturbation counterfactuals; limited real-world user studies for “plausibility.”
(regarding format: Related Work is in Appendix A.2, not a proper main-text section)

**Questions:**

- How sensitive are results to the attack method? It would be interesting to replace GOttack with random-orbit, k-hop heuristics, or Nettack-style candidates and report deltas.
- Why optimize an implausibility loss instead of pruning edits that violate constraints? It would be interesting to compare both.
- Please report full metrics for each dataset (not only average ranks) for all κ values,, and include confidence intervals where missing.
- Section 3 cites structural evasion. Why does the paper focus on it specifically? It would be interesting to discuss other attack types and whether conclusions change.

---

> ### Author Response · Authors · 2025-11-21
> **Reviewer GcK9: W1, W2**
>
> We thank the reviewer for taking the time to assess our paper and for the constructive suggestions. We appreciate the reviewer’s recognition of our originality, solid optimization mechanics, robustness checks, and the comprehensiveness of our study through large-scale, rigorous experimentation. Below, we address each of the reviewer’s comments in detail and have revised our manuscript accordingly, with changes highlighted in light blue.
>
> > W1: “Theory: “Hypothesis 1” is asserted as “proved” in the appendix but functions more as an assumption tied to empirical overlap; this weakens guarantees.”
>
> **Response**: Thank you for highlighting the theoretical positioning of Hypothesis 1. We appreciate the reviewer’s point and have revised the wording to avoid overstating Hypothesis 1. It is not presented as a formal guarantee, but as a model-driven hypothesis supported by gradient-based reasoning and consistent empirical overlap (Appendix A.12, Table 17). The hypothesis provides a useful lens for understanding how high-influence edge additions can align with counterfactual explanations. We have clarified this distinction in Section 3 with the following text:
>
> _“More importantly, we support this hypothesis with gradient-based reasoning and empirical similarity measures in Appendix A.12. To the best of our knowledge, this hypothesis and supporting evidence are presented for the first time as a plausible and empirically grounded link between adversarial perturbations and counterfactual explanations in graph learning.”_
>
> ---
>
> > W2: “Plausibility metric is coarse: degree/clustering penalties may not capture domain constraints like temporal or type compatibility”
>
> **Response**: Thank you for raising this important point regarding the limitations of general-purpose plausibility metrics and the need for domain-aware constraints. We would like to note that degree and motif constraints that we use are task agnostic, hence they help us achieve our good results. However, we agree that penalties based on them serve only as coarse structural proxies and may not fully capture context-specific constraints such as temporal validity, node-type compatibility, or semantic coherence.
>
> To address this, we have expanded the discussion in Appendix A.1 to outline how richer, domain-specific constraints (e.g., disallowing backward citation links, enforcing node-type compatibility) could be integrated into the ATEX-CF framework. These could take the form of hard constraints during candidate generation or additional soft losses during optimization.
>
> Moreover, we highlight that our **pruning step (Algorithm 2)** acts as an important post-hoc safeguard: by removing non-essential edges, it reduces the chance of retaining spurious or semantically implausible perturbations. As shown in Appendix A.8 (Figure 7), pruning discards unnecessary edits and sharpens the explanation to its minimal, most plausible core.
>
> Please note that an earlier review requested this as well, so we have colored the text orange in the PDF with the review order in mind.
>
> ---
>
> > W2b:  evaluation reuses the same surrogate.
>
> **Response**: Regarding the concern about surrogate reuse, we clarify that our evaluation does not rely on a single model. In addition to GCN, we include experiments with GAT and a Graph Transformer to verify that our fidelity and plausibility results generalize across architectures. These experiments (Appendix A.7, Tables 12–13) demonstrate that the effectiveness of ATEX-CF is robust to the choice of surrogate model.

---

> ### Author Response · Authors · 2025-11-21
> **Reviewer GcK9: W3, Q1, W4**
>
> > W3: “Dependence on attack generator: Candidate additions come from GOttack; robustness to this choice and ablations vs. simpler heuristics are unclear.”
>
> > Q1: “How sensitive are results to the attack method? Try random, k-hop, Nettack.”
>
> **Response**: Thank you for the suggestion. We conducted the requested ablations by replacing GOttack-based candidates with random sampling and Nettack-based selection.
>
> Table: Ablation study for attack candidate generation strategy on Cora and GCN
> |Method  | Base GNN  | Misclass. | Fidelity | ∆E (E+,E-) | Plausibility | Time(sec) |
> |------------|-------|--------|----------|--------|--------|-----------|
> | ATEX-CF+Random Sampling | GCN | 0.46 | 0.17 | 1.71 (0.74,0.97) | 0.67 | 7.44 |
> | ATEX-CF+Nettack                  | GCN | 0.67 | 0.22 | 1.62 (0.75,0.87) | 0.72 | 8.97 |
> | ATEX-CF+GOttack(ours)        | GCN | 0.69 | 0.22 | 1.66 (0.76,0.90) | 0.72 | 4.71 |
>
> As reported in a new Table 15 (Appendix A.9), attack-guided candidates yield substantially higher flip rates (≈69% improvement). This demonstrates that adversarial insights meaningfully enhance the effectiveness of candidate generation. Additionally, using GOttack candidates yields the highest misclassification rate and fidelity under the same budget, outperforming both random and Nettack-based baselines.
>
> Please note that an earlier review requested this as well, so we have colored the text red in the PDF with the review order in mind.
>
> ---
>
> > W4: “Scope: Only node classification; no feature-perturbation counterfactuals; limited real-world user studies for “plausibility.” (regarding format: Related Work is in Appendix A.2, not a proper main-text section)”
>
> **Response**: We thank the reviewer. Our current study focuses on structural counterfactual explanations for node classification tasks, where edge-level edits form the primary mode of intervention. We deliberately limited our focus because this is a seminal paper, and we would first like to establish the connection between counterfactuals and attacks. Extending ATEX-CF to feature perturbations and other tasks (e.g., graph classification, link prediction) is a natural next step, and we plan to incorporate node feature editing under plausibility constraints in future work.
>
> Regarding plausibility, we agree that real-world user evaluation is ultimately necessary. While our plausibility loss is based on graph-theoretic priors (e.g., degree and motif regularity), we view this as a proxy for real-world edits. We now clarify and discuss limitations more explicitly in Appendix A.1.

---

> ### Author Response · Authors · 2025-11-21
> **Reviewer GcK9: Q2, Q3, Q4**
>
> > Q2: “Why optimize an implausibility loss instead of pruning edits that violate constraints? Compare both.”
>
> **Response**: Thank you for the insightful suggestion. We believe that optimizing the loss directly is preferable to pruning for two reasons. First, the loss computation is efficient and integrates naturally into gradient-based optimization. Second, pruning after proposing a perturbation can be counterproductive in graph settings, where a single edge change can change the receptive fields of many nodes.
>
> We illustrate the second point with an example. Suppose that the model proposes the sequence of edits: $+e_7$, $+e_8$, $-e_2$. If $+e_7$ is deemed implausible and later pruned, the remaining edits $+e_8$ and $-e_2$ may no longer have any effect, as their impact depended on $+e_7$ being applied. This leads to wasted computation and potentially invalid counterfactuals. In contrast, our loss-based optimization penalizes implausible edits during training. As a result, $+e_7$ would be discarded early, and the model would avoid proposing follow-up edits that rely on its presence. This allows us to generate counterfactuals more efficiently and with greater consistency.
>
> In the first reason, we noted that plausibility loss is computationally cheap. This is because ATEX-CF restricts all computations to the ((l+1)-hop subgraph) of the target node (e.g., GCN’s 3 layers + 1), the maximum degree (d_max) is naturally bounded by this local neighborhood.
>
> Under this setting, the plausibility loss remains computationally lightweight. Even in worst-case large graphs such as ogbn-arxiv, the 4-hop neighborhood typically has (d_max≈50, n_sub≈1000), amounting to (1000 * 50^2 = 2.5*10^6) operations, still negligible compared to a single GNN forward pass O(L*|E|d + L * n_sub * d^2).  Other datasets’ numbers are given below.
>
>
>
>
>
> | Dataset | Global avg degree | Global max | Typical 4-hop local max | Reasonable d_max| Typical n_sub |
> | ---------- | ----------------- | ---------- | ----------------------- | ---------------- | ---------------- |
> | BA-SHAPES  | ~4        | <10        | 4–6                     | 5                  | 50-100         |
> | Cora              | ~3.7      | ~40        | 12–16                 | 15                | 100-300       |
> | ogbn-arxiv     | ~13       | ~1800    | 40–60                 | 50                | 500-1000     |
>
> Pruning is costly because it requires a GNN forward pass to detect which edges to perturb. Integrating the plausibility loss helps us avoid unnecessary, costly passes for candidates that would be pruned at the end anyway.
>
> We have added this discussion to  Appendix A.11.
>
> ---
>
> > Q3: “Report full metrics for each dataset (not only average ranks) for all κ values, with confidence intervals.”
>
> **Response**: We have added full tables for all datasets and all κ values, including confidence intervals, in Appendix A.6 (Tables 7–11).
>
> ---
>
> > Q4: “Why focus on structural evasion? Discuss other attack types and relevance.”
>
> **Response**: We focus on structural evasion attacks because they directly manipulate the graph topology, which is central to how GNNs aggregate information. Structural edits such as edge additions and deletions have immediate and interpretable effects on message passing and can be directly mapped to counterfactual graph modifications. This makes them particularly suitable for generating explanations that are both actionable and sparse.
>
> Other attack types, such as poisoning or feature perturbations, operate under different assumptions. Poisoning attacks require retraining the model after graph modification, which is incompatible with our setting, where the model remains fixed. Feature perturbations, while meaningful in some domains, often lack a structural grounding in graph tasks and are harder to constrain plausibly, e.g., editing feature vectors may violate domain semantics or cause cascading inconsistencies.
>
> By contrast, structural evasion attacks yield perturbations that naturally align with our goal: identifying minimal and plausible structural changes that flip model decisions. We have clarified this rationale in Section 2.3 and discuss potential extensions to other attack modes in Appendix A.1.
>
>
>
> “In this work, we focus on structural evasion attacks, which directly modify the graph topology at inference time. These attacks are particularly suited to our counterfactual framework, as they avoid retraining and yield interpretable perturbations that align with our goals of sparsity and plausibility.”
>
> “Our current framework does not consider poisoning attacks or node feature perturbations. Poisoning requires retraining the model after each perturbation, which conflicts with our fixed-model counterfactual setup. Feature perturbations, while useful in some domains, are harder to constrain plausibly and often lack structural interpretability in graph settings. Extending ATEX-CF to incorporate these forms of attacks is a promising direction for future work.”

---

> ### Author Response · Authors · 2025-11-24
> **Additional responses**
>
> Thank you once again for your thoughtful review.
>
> We have uploaded the latest version of our manuscript to OpenReview, where we have carefully addressed all of your previous concerns. In addition to the responses provided above, the revised submission includes the following updates:
>
> 1. Our full results are added to Tables 3 and 7–14 in Appendix A.6 - A.7. We now include evaluations against two recent counterfactual explanation approaches that support edge additions—INDUCE (Verma et al., 2024) and C2Explainer (Ma et al., 2025). Both methods allow edge additions and deletions, making them directly comparable to ATEX-CF. Descriptions of these methods are provided in Section 5.1, and comprehensive results across all datasets are reported in Tables 3 and 7–14. (Q3)
>
> We hope these revisions resolve any remaining questions and respectfully request your reevaluation in light of the additional evidence. We believe that your support for our paper will increase accordingly.

---

### Official Review · Reviewer_wuaS · 2025-10-31

**Soundness:** 2
**Presentation:** 3
**Contribution:** 3
**Rating:** 4
**Confidence:** 4

**Summary:**

This paper claims that existing counter-factual (CF) explanation methods for GNNs mainly focus on edge removing, while largely overlooking edge adding as a perturbation method. They explained (with examples) why edge adding is also important in counter-factual explanation, and also showed that GNN attack methods often employ edge adding as a perturbation method. Motivated by this, they combine edge removing (in conventional CF-Explaination methods) with edge adding (in GNN attack scenarios) to obtain a new method for GNN CF-explanation. Next, they perform experiments to study the effectiveness. While the idea looks interesting, I have some concerns regarding some claims and the experiment studies (please the weak points).

**Strengths:**

1. The idea is interesting. While some recent work also considered edge adding, it is still novel to use GNN attack as a source of obtaining edges to add;
2. The paper is easy to follow as they spent many efforts to motivate their research and explain their methodology. However, the space left for the experiments looks relatively too short (this can be improved);
3. The related work section (though in the appendix) is very detailed.

**Weaknesses:**

The experiments are not convincing due to two facts:

1. They considered very limited baselines. Particularly, as a CF-explanation method, they only considered one method (CF-GNNExplainer) as a baseline. However, there are so many other CF-explanation methods (they also reviewed these methods in related work), including RCExplainer, GNN-MOExp, CF$^2$, NSEG, Banzhaf, CF-GFExplainer; INDUCE, C2Explainer, CLEAR, GCFExplainer, etc. (Actions: add more baselines in this category)

2. The datasets considered are limited and may be flawed. They considered only five datasets, including BA-SHAPES, TREE-CYCLES, Loan-Decision, Cora, ogbn-arxiv. Besides, as pointed in [1], the explanations given in these datasets might not be reliable, using wrong/unreliable ground-truth to evaluate GNN explanation methods is not convincing [2]. (Actions: Discussion on potential issues of these datasets, adding more real or synthetic datasets)

Besides, the authors claim that ``ATEX-CF generates explanations that are not only faithful but also informative." Without solving these potential flaws in the dataset, it is hard to say whether the explanations are "faithful". Moreover, without a real use-case, it is hard to judge whether the explanation is "informative". (Actions: investigating and solving/mitigating flaws in these datasets, providing a use-case)

Moreover, although the authors have considered using plausibility loss to prevent generating out-of-distribution explanations, the OOD problem is actually very domain-specific. Avoiding implausible degree jumps and implausible clustering jumps (namely these techniques used in the paper) can only slightly mitigate these issues. (Actions: I would like to see more discussions regarding this and maybe other solutions to avoid generating OOD explanations).


## Reference

[1] Agarwal, Chirag, Owen Queen, Himabindu Lakkaraju, and Marinka Zitnik. "Evaluating explainability for graph neural networks." Scientific Data 10, no. 1 (2023): 144.

[2] Faber, Lukas, Amin K. Moghaddam, and Roger Wattenhofer. "When comparing to ground truth is wrong: On evaluating gnn explanation methods." In Proceedings of the 27th ACM SIGKDD conference on knowledge discovery & data mining, pp. 332-341. 2021.

**Questions:**

Please see the weak points.

---

> ### Author Response · Authors · 2025-11-21
> **Part 1**
>
> We thank the reviewer for taking the time to review our paper and for offering valuable suggestions. We are pleased that the reviewer describes our experimental setup as a significant contribution to the community. In the following, we will reply to the reviewer's questions and comments and revise our manuscripts to incorporate the reviewer’s feedback in orange.
>
> ## W1
> > “Baselines are too limited; many CF-explanation methods are missing (RCExplainer, GNN-MOExp, INDUCE, C2Explainer, etc.).”
>
> **Response**: Thank you for pointing out the opportunity to broaden our baseline coverage. Due to time limitations, we have completed the run of INDUCE (Verma et al., 2024) and C2Explainer (Ma et al., 2025) on the CORA dataset, which we show in the table below. Results from other datasets will be shared as they become available before the discussion deadline.
>
> The results show that ATEX-CF continues to outperform these methods in misclassification rate and plausibility. In plausibility, ATEX-CF outperforms INDUCE and ranks similar to C2Explainer (0.75 vs. 0.68). C2Explainer is computationally costlier than ATEX-CF, and its misclassification is much smaller than ATEX-CF (0.61 vs. 0.72).
>
> Table: Performance of counterfactual explanations on Cora and GCN
> | Method | Base GNN | Misclass. | Fidelity | ∆E(E+,E-) | Plausibility | Time(sec) |
> |-------------------|--------------------|--------------|-----------|------------|----------------|--------|
> | CF-GNNExplainer | GCN | 0.49  | 0.11    | 1.70(0.00,1.70)  | 0.64  | 10.2 |
> | INDUCE                | GCN | 0.17  | 0.03    | 2.66(1.50,1.16)  | 0.27  | 0.9 |
> | C2Explainer          | GCN | 0.60  | 0.21    | 2.60(0.00,2.60)  | **0.78**  | 9.5 |
> | GNNExplainer       | GCN | 0.22  | 0.02    | 2.58(0.00,2.58)  | 0.53  | 0.4 |
> | PGExplainer          | GCN | 0.14  | 0.00   | 2.38(0.00,2.38)   | 0.53  | **0.1** |
> | Nettack                  | GCN | 0.53  | 0.15    | 5.00(3.86,1.14)  | 0.13  | 3.4 |
> | GOttack                 | GCN | 0.53  | 0.15    | 5.00(4.70,0.30)  | 0.10  | 2.2 |
> | ATEX-CF(ours)      | GCN | **0.72**  | **0.23**    | **1.63(0.90,0.73)**  | 0.75  | 7.3 |

---

> ### Author Response · Authors · 2025-11-21
> **Part 2**
>
> ## W2
> >  “The datasets considered are limited and may be flawed. They considered only five datasets, including BA-SHAPES, TREE-CYCLES, Loan-Decision, Cora, ogbn-arxiv. Besides, as pointed out in [1], the explanations given in these datasets might not be reliable; using wrong/unreliable ground-truth to evaluate GNN explanation methods is not convincing [2]. (Actions: Discussion on potential issues of these datasets, adding more real or synthetic datasets)”
>
> **Response**: Thank you for your insightful comments regarding the datasets and evaluation methods in our paper.
>
> We clarify that ATEX-CF is a counterfactual explanation method, whose goal is not to match predefined ground-truth motifs (as in factual explainers) but to identify minimal, plausible edits that flip the model’s prediction. Therefore, as emphasized by prior work (Agarwal et al., Faber et al.), ground-truth–overlap metrics are inappropriate here; counterfactuals should instead be evaluated by faithfulness (fidelity for counterfactuals - prediction probability drops), minimality, and plausibility, which is exactly how we assess ATEX-CF.
>
> We have added the following paragraph to the related work:
>
> _**Pitfalls in Explanations**_
>
> _Faber et al. identify five pitfalls that affect the evaluation of GNN explanation methods: (1) the GNN may rely on bias terms or spurious features rather than the annotated evidence, (2) the ground-truth explanation may be redundant or non-unique, leading to mismatches during scoring, (3) some datasets allow trivial explanations, such as those based on nearest neighbors or centrality, (4) weak models that do not learn the true structure render all explanation assessments unreliable, and (5) explanation behavior may vary significantly across architectures even with similar accuracy. These concerns are valid for attribution-based explanations evaluated against fixed motifs, but they do not apply to our setting. Our method generates counterfactuals by identifying sparse edge perturbations that flip the model’s prediction. We do not assume or require ground-truth substructures, nor do we compare explanations to predefined motifs. Instead, our evaluation reflects the model’s actual behavior and is based on plausibility and edit sparsity. This makes our approach robust to the ground-truth mismatches discussed in~[31,12]._
>
> Furthermore, **we have conducted experiments on a new real-world (heterophilic) Chameleon dataset**. The table below demonstrates the performance of our counterfactual explanations on this additional real-world dataset compared to the baselines.
>
> Table: Performance of counterfactual explanations on Chameleon and GCN.
> | Method | Base GNN | Misclass. | Fidelity | ∆E(E+,E-) | Plausibility | Time(sec) |
> |-------------------|--------------------|--------------|-----------|------------|----------------|--------|
> | CF-GNNExplainer | GCN |   |  |  |  |  |--ok
> | INDUCE                | GCN |  |  |  |  |  |--ok
> | C2Explainer          | GCN | 0.09 | 0.06 | 2.61(2.17,0.44)  | **0.76**  | 86.0 |
> | GNNExplainer       | GCN | 0.01 | 0.01 | 3.00(0.00,3.00)  | 0.55  | 11.2 |
> | PGExplainer          | GCN |  |  |  |  |  |--ok
> | Nettack                  | GCN | 0.52 | 0.18 | 5.00(4.73,0.27)  | 0.29  | 38.6 |
> | GOttack                 | GCN | 0.52 | 0.19 | 5.00(4.87,0.13)  | 0.43  | **9.3** |
> | ATEX-CF(ours)      | GCN | **0.84** | **0.26** | 1.64(1.16,0.48)  | **0.76**  | 134.7 |
>
> The missing results of the table will be completed by the discussion dealine.

---

> ### Author Response · Authors · 2025-11-21
> **Part 3**
>
> ## W3
> >  “Claims about faithfulness and informativeness are hard to justify without better datasets or a real use-case. (Actions: investigating and solving/mitigating flaws in these datasets, providing a use-case)”
>
> **Response**: Thank you for the suggestion. We have created a case study of an attack on node 1978 of the CORA dataset, with a classification explanation and its counterfactual edits identified by our model. We are not able to share the resulting counteractual Figure 7 of the PDF on OpenReview, but we have added the figure and its discussion to Appendix A.8 with the following text:
>
> _“To illustrate the refinement effect of pruning, Figure~7 compares the initial and final counterfactual explanations for a target node. Before the pruning, the explanation includes several edges of varying influence, some of which are unnecessary for achieving the prediction flip. The attack procedure discards these redundant edits by greedily testing their necessity. In this case, the final counterfactual achieves its goal by deleting the edge $(1978, 1306)$ and adding a single edge $(1978, 190)$. This minimal perturbation is sufficient to alter the model's decision.”_

---

> > ### Author Response · Authors · 2025-11-21
> > **Part 4**
> >
> > ## W4
> > >  “Although the authors have considered using plausibility loss to prevent generating out-of-distribution explanations, the OOD problem is actually very domain-specific. Avoiding implausible degree jumps and implausible clustering jumps (namely these techniques used in the paper) can only slightly mitigate these issues. (Actions: I would like to see more discussions regarding this and maybe other solutions to avoid generating OOD explanations)”
> >
> > **Response**: Thank you for this insightful comment. We agree that the OOD nature depends on domain-specific factors, and generic constraints alone may not suffice. In our framework, we use two structure-aware losses  (degree anomaly and motif violation) to prevent perturbations that cause abnormal topological changes. However, as you rightly point out, these constraints act as soft regularizers and may not be enough for factors like temporal plausibility.
> >
> > To address this, we have expanded our discussion in Appendix A.1 to explicitly acknowledge the limitations of our plausibility model and the potential need for domain-specific rules or learned constraints. Moreover, beyond the training-time plausibility loss, we would like to point out that our pruning step (Algorithm 2) acts as a filter against spurious edits. Since pruning removes any edge that is not necessary for prediction flip, it tends to eliminate low-importance or noisy perturbations, including those that may be implausible. For instance, in the case study visualized in Figure 7 (Appendix A.8), the initial explanation includes several important edges, but the final minimal solution retains only one addition and one deletion, both semantically aligned with the target class.
> >
> > We view this late pruning as an important complement to the plausibility loss. While it does not explicitly model domain constraints, it reduces the number of edits and therefore the opportunity for OOD behavior to emerge. In future work, we plan to incorporate domain-aware constraints more directly, such as via graph embeddings, node-type priors, or rule-based filters that restrict candidate perturbations during search.
> >
> > Added text in Section A1: _“We also recognize that the current plausibility loss, based on structural proxies such as degree anomaly and motif violations, may only partially address the risk of generating out-of-distribution explanations. These generic regularizers do not account for domain-specific constraints such as temporal consistency or semantic incompatibility (e.g., citing future papers or linking unrelated functional modules). To mitigate this, we emphasize that our pruning step (Algorithm~2) serves as a late filter that discards non-essential edits, including those that may be structurally valid yet semantically implausible. In practice, pruning significantly reduces the number of active perturbations, helping ensure that final explanations are not only minimal but more likely to remain within the distribution.”_

---

> ### Author Response · Authors · 2025-11-24
> **Additional responses**
>
> Thank you once again for your constructive review.
>
> We have now uploaded the latest version of our pdf to OpenReview and have carefully addressed all your past concerns.
>
> Specifically, in addition to our responses above, in the new pdf, we have added:
>
> 1. Additional comparisons with recent methods: We now include evaluations against two recent counterfactual explanation approaches that support edge additions—INDUCE (Verma et al., 2024) and C2Explainer (Ma et al., 2025). Both methods allow edge additions and deletions, making them directly comparable to ATEX-CF. Descriptions of these methods are provided in Section 5.1, and comprehensive results across all datasets are reported in Tables 3 and 7–14. (W3 & Q1)
>
> 2. To further assess generalization beyond homophilic benchmarks, we have added a heterophilic Chameleon dataset in Section 5.1 and Table 2. We provide complete results (Table 14)  for all baselines and analysis (Appendix A.6) on the Chameleon dataset. (W2)
>
> We hope that these revisions resolve any remaining questions and respectfully request your reevaluation in light of the additional evidence.

---

> > ### Comment · Reviewer_wuaS · 2025-11-27
> >
> > Dear Authors,
> >
> > I just carefully read the detailed responses, which have partially addressed my concerns. Due to limited time, I understand it is challenging to add more experiments (ecpeically more baselines). However, the claims can be largely strengthened if more baselines of the same category are considered. Therefore, I will raise my overall rating (but not too much) from 4 to 6. Thank you for your contributions.

---

> > > ### Author Response · Authors · 2025-11-30
> > > **Reviewer wuaS; Additional baselines added**
> > >
> > > >  “I just carefully read the detailed responses, which have partially addressed my concerns. Due to limited time, I understand it is challenging to add more experiments (espeically more baselines). However, the claims can be largely strengthened if more baselines of the same category are considered. ”
> > >
> > > **Response:**
> > >
> > > Thank you very much for taking the time to re-evaluate our responses, increasing your score (from 4 to 6) and for your additional constructive comments.
> > >
> > > We fully agree that incorporating additional baselines of the same category can further strengthen the empirical claims. In response, we have included two more counterfactual explanation baselines—CF2 and NSEG—and updated the experimental results accordingly (Table 3 in Section 5.2 and Tables 7–14 in Appendix A.6–A.7). The new results consistently demonstrate that ATEX-CF maintains superior flip rates and plausibility across multiple datasets, further reinforcing the effectiveness of our method.
> > >
> > > Thank you again for your insightful comments, which helped us improve the completeness and rigor of the experimental evaluation.

---

### Official Review · Reviewer_7BBe · 2025-11-02

**Soundness:** 3
**Presentation:** 3
**Contribution:** 3
**Rating:** 6
**Confidence:** 3

**Summary:**

This paper addresses a key limitation in GNN counterfactual (CF) explanations, CF methods relying only on edge deletion often cannot find a counterfactual, especially when the prediction hinges on the absence of certain edges. The authors propose ATEX-CF, a hybrid framework that intelligently combines edge additions and deletions. It first uses an adversarial attack (GOttack) to identify impactful candidate additions and then jointly optimizes both additions and deletions. It also prunes the resulting edits for minimality. The authors theoretically prove situations where deletions fail , and empirically show deletion only methods frequently fail on some datasets. Across synthetic and real world benchmarks (like Cora and ogbn-arxiv), ATEX-CF demonstrated higher flip rates with fewer, more plausible edits than existing CF and attack baseline.

**Strengths:**

1. Strong theoretical and empirical evidence justifies including edge additions for counterfactual explainability.
2. The attack guided candidate generation is innovative; the signed mask, STE, and pruning pipeline coherently addresses both search efficiency and plausibility.
3. ATEX-CF achieves higher flip rates using fewer, more plausible edits across various datasets and GNN models.
4. The work is framed as the first to bridge GNN adversarial attacks with counterfactual explanations, distinguishing it from prior deletion only or unguided addition methods.

**Weaknesses:**

1. The pipeline is complex and densely explained.
2. The  additions and deletions are assumed to be equally feasible, asymmetric costs are not explored.
3. Evaluation misses some recent counterfactual baselines that also permit additions, such as (InduCE: InduCE: Inductive Counterfactual Explanations for Graph Neural Networks | OpenReview ), slightly weakening the claim of empirical dominance.
4. Released GitHub code is difficult for international researchers to reproduce, as some comments are in different languages other than english.
5. Novelty claims (bridging attacks and CFs) are specific to GNNs; this conceptual link was explored earlier in other domains (e.g., The Intriguing Relation Between Counterfactual Explanations and Adversarial Examples | Minds and Machines  ).

**Questions:**

* Could you compare against recent non attack guided CF explainers that also use edge additions to isolate the benefit of the “attack informed” step?
* It would be nice to ablate the candidate generation strategy (e.g., replacing attack guided edges with simpler heuristics such as random sampling, degree based, or feature similarity based candidates) to quantify how much this  guidance  step contributes to the final performance.
* Can the framework handle domains where additions are infeasible or more expensive than deletions? Have you tried asymmetric action costs?

---

> ### Author Response · Authors · 2025-11-21
> **Part 1**
>
> We thank the reviewer for taking the time to review our paper and for offering valuable suggestions. Here, we address each point raised by the reviewer as follows and revise our manuscript to incorporate the reviewer’s feedback in red in the updated PDF.
>
>
> ### W1
> >  “The pipeline is complex and densely explained.”
>
> **Response**: We thank the reviewer for the comment. We have revised the text to clarify the pipeline. In Section 4.2, we now refer to Algorithm 1 and Algorithm 2 step by step, present pseudocode for the full ATEX-CF workflow, and make all procedural steps explicit. Algorithm~2 now includes a detailed description of the pruning mechanism in Section A4. In the main text, we link key lines of the algorithms (both Algorithms 1 and 2) to their explanations to help readers follow the workflow more easily.

---

> ### Author Response · Authors · 2025-11-21
> **Part 2**
>
> ### W2 and Q3
> > W2: “Additions and deletions are assumed equally feasible; asymmetric costs not explored.”
>
> > Q3: “Can the framework handle domains where additions are infeasible or more expensive?”
>
> **Response**: Thanks for the suggestion. You raise an interesting point that we have not seen addressed in the literature yet; what happens if the budget is real valued with tailored costs for each perturbation type? For the perturbation budget constraint ||\delta A||0 =||\Delta E+|| + ||\Delta E-||≤κ, we changed the budget constraint to C*||\Delta E+|| + ||\Delta E-||≤κ by defining a given asymmetric cost parameter C. The results are shown in three tables below, where we adjust the cost of addition w.r.t. deletion in the range (0.5, 21.0) for three budgets (κ=5, 10, 20) in two tables below and the Table 4 of the PDF, respectively. If both addition and deletion are equally costly (e.g., row 3 of 1.0 and 1.0), the costs are symmetric. As the addition cost increases towards the budget, the sparsity objective would favor deletions.
>
> Table: Counterfactual explanations of asymmetric cost C on Cora and GCN under κ=5
> | Addition Cost | Deletion Cost | Misclass. | Fidelity | ∆E(E+,E-) | Plausibility | Time(sec) |
> |-------------------|--------------------|--------------|-----------|------------|----------------|--------|
> | 0.5 | 1.0 | 0.69  | 0.23    | 1.69(0.78,0.91)  | 0.72  | 10.5 |
> | 0.8 | 1.0 | 0.69  | 0.22    | 1.66(0.77,0.89)  | 0.72  | 10.4 |
> | 1.0 | 1.0 | 0.69  | 0.22    | 1.66(0.76,0.90)  | 0.72  | 10.2 |
> | 2.0 | 1.0 | 0.67  | 0.22    | 1.59(0.65,0.94)  | 0.71  | 10.5 |
> | 2.5 | 1.0 | 0.64  | 0.20    | 1.49(0.62,0.87)  | 0.71  | 10.7 |
> | 5.0 | 1.0 | 0.53  | 0.15    | 1.35(0.49,0.86)  | 0.68  | 11.2 |
> | 6.0 | 1.0 | 0.43  | 0.10    | 1.59(0.00,1.59)  | 0.61  | 11.3 |
>
>
> Table: Counterfactual explanations of asymmetric cost C on Cora and GCN under κ=10
> | Addition Cost | Deletion Cost | Misclass. | Fidelity | ∆E(E+,E-) | Plausibility | Time(sec) |
> |-------------------|--------------------|--------------|-----------|------------|----------------|--------|
> | 0.5 | 1.0 | 0.70  | 0.23    | 1.78(0.78,1.00)  | 0.72  | 5.9 |
> | 0.8 | 1.0 | 0.70  | 0.23    | 1.78(0.77,1.01)  | 0.71  | 6.0 |
> | 1.0 | 1.0 | 0.70  | 0.23    | 1.78(0.77,1.02)  | 0.71  | 7.4 |
> | 3.0 | 1.0 | 0.70  | 0.23    | 1.77(0.66,1.11)  | 0.69  | 7.7 |
> | 5.0 | 1.0 | 0.68  | 0.22    | 1.71(0.61,1.10)  | 0.69  | 7.8 |
> | 7.0 | 1.0 | 0.68  | 0.22    | 1.70(0.59,1.10)  | 0.69  | 7.8 |
> | 10  | 1.0 | 0.54  | 0.15    | 1.42(0.49,0.93)  | 0.68  | 8.2 |
> | 11  | 1.0 | 0.42  | 0.10    | 1.78(0.00,1.78)  | 0.62  | 8.5 |
>
> Table 4: Counterfactual explanations of asymmetric cost C on Cora and GCN under κ=20
> | Addition Cost | Deletion Cost | Misclass. | Fidelity | ∆E(E+,E-) | Plausibility | Time(sec) |
> |-------------------|--------------------|--------------|-----------|------------|----------------|--------|
> | 0.5 | 1.0 | 0.70  | 0.23    | 1.78(0.78,1.00)  | 0.72  | 6.1 |
> | 0.8 | 1.0 | 0.70  | 0.23    | 1.78(0.77,1.01)  | 0.71  | 6.1 |
> | 1.0 | 1.0 | 0.70  | 0.23    | 1.78(0.77,1.01)  | 0.71  | 10.3 |
> | 3.0 | 1.0 | 0.70  | 0.23    | 1.82(0.66,1.16)  | 0.69  | 10.7 |
> | 5.0 | 1.0 | 0.70  | 0.23    | 1.82(0.65,1.17)  | 0.69  | 10.9 |
> | 10  | 1.0 | 0.69  | 0.23    | 1.80(0.61,1.19)  | 0.69  | 11.0 |
> | 15  | 1.0 | 0.69  | 0.23    | 1.76(0.60,1.16)  | 0.69  | 11.0 |
> | 20  | 1.0 | 0.54  | 0.15    | 1.42(0.49,0.93)  | 0.68  | 11.5 |
> | 21  | 1.0 | 0.42  | 0.10    | 1.78(0.00,1.78)  | 0.62  | 11.7 |
>
> **Analysis of Asymmetric Weight Effects**: These results show that increasing the asymmetric addition cost C gradually shifts the perturbations from mixed edge additions/deletions toward deletion-dominant or deletion-only solutions. As C grows, the model relies less on expensive additions, leading to decreased misclassification ability (i.e., the ability to successfully flipping the prediction) and decreased fidelity, as well as slightly lower plausibility. This confirms that asymmetric weighting effectively controls the balance between addition and deletion operations, whereas overly large C values (e.g., C≥5) overly restrict additions (delete-only strategy) and degrade counterfactual quality.
>
> This discussion is added as a new Section 5.3 in the PDF with Table 4 for κ=20.

---

> > ### Author Response · Authors · 2025-11-21
> > **Part 3**
> >
> > ### W3 and Q1
> > > W3: “Evaluation misses recent baselines that support edge additions (e.g., INDUCE).”
> > > Q1: “Could you compare against recent non-attack CF explainers that also use additions?”
> >
> > **Response**: We appreciate the reviewer pointing this out. Due to time limitations, we have completed the run of INDUCE (Verma et al., 2024) and C2Explainer (Ma et al., 2025) on the CORA dataset, which we show in the table below. The results of other datasets will be shared as they arrive before the discussion deadline.
> >
> > The results show that ATEX-CF continues to outperform these methods in misclassification rate and plausibility. In plausibility, ATEX-CF outperforms INDUCE and ranks similarly to C2Explainer (0.75 vs. 0.78). C2Explainer performs well; however, it is computationally costlier than ATEX-CF, and its misclassification is much smaller than ATEX-CF (0.6 vs. 0.72).
> >
> > Table: Performance of counterfactual explanations on Cora and GCN
> > | Method | Base GNN | Misclass. | Fidelity | ∆E(E+,E-) | Plausibility | Time(sec) |
> > |-------------------|--------------------|--------------|-----------|------------|----------------|--------|
> > | CF-GNNExplainer | GCN | 0.49  | 0.11    | 1.70(0.00,1.70)  | 0.64  | 10.2 |
> > | INDUCE                | GCN | 0.17  | 0.03    | 2.66(1.50,1.16)  | 0.27  | 0.9 |
> > | C2Explainer          | GCN | 0.60  | 0.21    | 2.60(0.00,2.60)  | **0.78**  | 9.5 |
> > | GNNExplainer       | GCN | 0.22  | 0.02    | 2.58(0.00,2.58)  | 0.53  | 0.4 |
> > | PGExplainer          | GCN | 0.14  | 0.00   | 2.38(0.00,2.38)   | 0.53  | **0.1** |
> > | Nettack                  | GCN | 0.53  | 0.15    | 5.00(3.86,1.14)  | 0.13  | 3.4 |
> > | GOttack                 | GCN | 0.53  | 0.15    | 5.00(4.70,0.30)  | 0.10  | 2.2 |
> > | ATEX-CF(ours)      | GCN | **0.72**  | **0.23**    | **1.63(0.90,0.73)**  | 0.75  | 7.3 |

---

> ### Author Response · Authors · 2025-11-21
> **Part 4**
>
> ## W4
> >  “Released GitHub code includes non-English comments, making reproduction difficult.”
>
> **Response**: We apologize for this oversight. All comments in the repository have now been translated into English, and additional documentation has been provided to improve reproducibility.
>
> ---
>
> ## W5
> > “Novelty claims (bridging attacks and CFs) were explored earlier in other domains. (e.g., The Intriguing Relation Between Counterfactual Explanations and Adversarial Examples | Minds and Machines )”
>
> **Response**: Thank you for noting the interesting prior work; we were not aware of it.  The article studies how counterfactual explanations and adversarial examples relate in standard machine learning models, including their historical development and use cases. In fact, the author reaches the conclusion we stated: that counter- and adversarial examples can be obtained by solving the same optimization problem.  However, the author considers image and tabular data settings, but applying the framework to GNNs requires modifications.
>
> Our contribution lies in formalizing and adapting this connection to GNNs, where structural dependencies, such as directed edges, introduce unique challenges. We now cite the reference prominently in the introduction, where we motivate the problem. We have also added the following sentences to the related work section:
>
> _“We begin a complete account of counterfactual explanations and adversarial examples with their philosophical origins, where both have been studied as forms of contrastive reasoning and causal dependence._
>
> _Freiesleben  examines the connection between counterfactual explanations and adversarial examples in standard machine learning, studying their shared optimization goal and conceptual and historical development in fields such as philosophy and psychology. The analysis remains focused on models with independent input features, where perturbations do not propagate through structured dependencies. Our work builds on this by studying the connection in graph neural networks, where node predictions depend on message passing over edges. In this setting, identifying minimal changes that flip predictions becomes harder, and naive perturbations can easily break plausibility. We address these challenges by using adversarial edge additions to guide counterfactual generation and show that attack-informed edits offer an effective and realistic way to produce explanations in graphs.”_
>
>  ---
>
> ## Q2
> >  “Ablate the candidate generation strategy (e.g., random, degree-based) to quantify the benefit of attack guidance.”
>
> **Response**: We conducted the requested ablations by replacing GOttack-based candidates with random sampling and Nettack-based selection.
>
> Table: Ablation study for attack candidate generation strategy on Cora and GCN
> |Method  | Base GNN  | Misclass. | Fidelity | ∆E(E+,E-) | Plausibility | Time(sec) |
> |------------|-------|--------|----------|--------|--------|-----------|
> | ATEX-CF+Random Sampling | GCN | 0.46 | 0.17 | 1.71(0.74,0.97) | 0.67 | 7.44 |
> | ATEX-CF+Nettack                  | GCN | 0.67 |**0.22** | 1.62(0.75,0.87) | **0.72** | 8.97 |
> | ATEX-CF+GOttack(ours)        | GCN | **0.69** | **0.22** | 1.66(0.76,0.90) | **0.72** | **4.71** |
>
> As reported in this table, attack-guided candidates yield substantially higher flip rates (≈69% improvement). This demonstrates that adversarial insights meaningfully enhance the effectiveness of candidate generation. Additionally, using GOttack candidates yields the highest misclassification rate and fidelity under the same budget, outperforming both random and Nettack-based baselines.

---

> ### Author Response · Authors · 2025-11-24
> **Additional responses**
>
> Thank you once again for your thoughtful review and constructive feedback.
>
> We have uploaded the latest version of our manuscript to OpenReview, where we have carefully addressed all of your previous concerns. In addition to the responses provided above, the revised submission includes the following updates:
>
> 1. Clarification of asymmetric cost results: We have expanded our discussion by adding Appendix A.13, which provides a more detailed explanation of the observed outcomes. (W2 & Q3)
>
> 2. Additional comparisons with recent methods: We now include evaluations against two recent counterfactual explanation approaches that support edge additions—INDUCE (Verma et al., 2024) and C2Explainer (Ma et al., 2025). Both methods allow edge additions and deletions, making them directly comparable to ATEX-CF. Descriptions of these methods are provided in Section 5.1, and comprehensive results across all datasets are reported in Tables 3 and 7–14. (W3 & Q1)
>
> We hope these revisions satisfactorily address your remaining questions and further strengthen your support for our work.

---

> > ### Comment · Reviewer_7BBe · 2025-11-24
> >
> > Dear authors,
> >
> > Thank you very much for your very detailed rebuttal! I've been reading all of your experiments and I must say that I'm much more convinced of the results you presented.
> >
> > I will raise my score.
> >
> > Thanks

---

> ### Author Response · Authors · 2025-11-25
>
> Dear Reviewer,
>
> Thank you very much for your thoughtful follow-up and for taking the time to read our additional experiments in detail. We truly appreciate your constructive feedback throughout the process and are grateful for your recognition of our work.
>
> Thank you again for helping improve the paper.
>
> Best regards,
>
> The authors

---

### Author Response · Authors · 2025-11-30
**Summary of discussion and updates**

Dear chairs,

We have revised the submission and addressed the reviewers’ comments with new experiments, clarifications, and additional analysis. In the updated PDF, responses are color-coded. Below, we summarize the status of the main concerns.

**Reviewer 7BBe (rating 6 to 8 after rebuttal)** asked for asymmetric perturbation costs, stronger evidence that attack guidance is beneficial beyond simpler heuristics, more recent counterfactual baselines that support edge additions, and better clarity on the pipeline and related conceptual links. We extended the budget constraint to a real-valued form with a cost parameter on additions, ran a full study on Cora for several budgets and cost values, and added a new section 5.3 as well as Appendix A.13 to show how a higher addition cost gradually shifts explanations toward deletion-only solutions (with reduced flip rates and fidelity). We added 4 new baselines, including INDUCE, C2Explainer, CF2, and NSEG, and showed on Cora and the other datasets that ATEX-CF retains higher flip rates while remaining competitive in plausibility. We provided ablations where Gottack-based candidates are replaced by random sampling or Nettack, and reported that attack-informed candidates give higher flip rates and fidelity under the same budget. After these updates, the reviewer stated that they were much more convinced and raised the score.

**Reviewer wuaS (rating 4 to 6)** was concerned about the breadth of baselines, the reliability of popular explanation datasets, the strength of faithfulness and informativeness claims, and the limits of our plausibility model. We broadened the set of counterfactual baselines by adding INDUCE and C2Explainer, and reported full comparisons, including a new heterophilic Chameleon benchmark to test generalization beyond the original five datasets. We clarified that ATEX-CF does not rely on  ground truth motifs and that we follow the view in recent work that counterfactual explanations should be evaluated by flip rate, sparsity, and plausibility rather than overlap with noisy "ground truth" motifs. We added a case study on a Cora node to show how pruning refines explanations to a minimal and interpretable pair of edits, and we expanded Appendix A.1 to state more clearly that our degree and motif-based plausibility losses are structural proxies, not a complete solution to out-of-distribution behavior, and that domain-specific rules would be a natural extension. These additions partly addressed the reviewer’s concerns, and they raised their score to 6. The reviewer further suggested adding more baselines, and we thereafter included two additional counterfactual baselines CF2 and NSEG, and showed on Cora and the other datasets that ATEX-CF still retains higher flip rates and plausibility.

**Reviewer GcK9 (rating currently 4)** focused on the theoretical status of our hypothesis connecting attacks and counterfactuals, the coarseness of the plausibility metric and reuse of a surrogate, the dependence on GOttack, and the restriction to structural evasion on node classification. We softened the language around Hypothesis 1, explicitly presenting it as a model-driven hypothesis supported by gradient reasoning and empirical overlap, rather than a formal guarantee. We expanded the plausibility discussion to explain how domain-specific constraints could be added, and we highlighted that pruning acts as a second filter that removes unnecessary or implausible edits. We clarified that our experiments include GCN, GAT, and a graph transformer to check robustness across architectures, and added full per-dataset tables with confidence intervals for all budgets. We also introduced ablations that replace GOttack with random and Nettack-based candidates and showed that GOttack yields the best flip rates and fidelity with lower runtime, which supports our attack-informed design. Finally, we explained why we focus on structural evasion in this first work and outlined how poisoning and feature perturbations could be incorporated later while keeping the model fixed.

**Reviewer M2c7 (rating 6)** viewed the novelty and quality of the work positively and mainly asked for a clearer discussion of limitations and future directions. We moved the discussion of the assumption that additions and deletions are always feasible from the appendix to the main paper and complemented it with the asymmetric cost study, which shows how to downweight or effectively forbid additions when needed. We also added a short forward-looking section describing extensions to dynamic graphs, feature perturbations, and applications in domains such as healthcare and finance.

---

> ### Author Response · Authors · 2025-11-30
> **Summary of discussion and updates (continued)**
>
> Across all reviews, no one has questioned the soundness of the method or the validity of our experiments. Two reviewers explicitly raised their scores after reading the rebuttal and new results, and the remaining concerns now center on scope and future extensions rather than correctness. We believe the new experiments, broader baselines, asymmetric cost study, and stronger discussion of plausibility and theory substantially strengthen the case for ATEX-CF as a robust and practical connection between adversarial attacks and counterfactual explanations for graph neural networks.

---

### Meta-Review · Area_Chair_WFhq · 2026-01-13

**Summary:**

The submission presents ATEX-CF, a framework that creates counterfactual explanations for GNNs by utilizing adversarial attack techniques, particularly guided edge adds. The fundamental idea is that more thorough explanations that go beyond straightforward edge removals are made possible by including attack-based insights.

Reviews were mixed at first (scores of 6, 4, 4, 6). The experimental rigor (absence of recent baselines), the theoretical framing of the relationship between attacks and counterfactuals, and the absence of analysis on asymmetric perturbation costs raised serious concerns, even though reviewers valued the novelty of bridging attacks and explanations.

The rebuttal stage was really fruitful. The writers produced a substantial amount of original work, such as:

1. **Benchmark:** Added comparisons against four recent baselines that support edge additions (INDUCE, C2Explainer, CF2, NSEG).
2. **Robustness:** Introduced a new heterophilic dataset (Chameleon) and performed ablation studies on the candidate generation strategy (Random vs. Nettack vs. GOttack).
3. **Sensitivity Analysis:** Addressed a key conceptual question regarding asymmetric costs for edge additions vs. deletions.

Following these updates, two reviewers explicitly raised their scores (one from 6 to 8, another from 4 to 6).

**Reviewer Concerns:**

**Addressed Concerns:**

- **Missing Baselines (wuaS, 7BBe):** The main criticism was that there was no comparison with cutting-edge techniques that permit edge additions. The authors show that ATEX-CF retains superior flip rates and plausibility scores by successfully integrating INDUCE and C2Explainer.
- **Asymmetric Costs (7BBe):** The reviewer questioned the notion that removals and additions are equally possible. In a thorough analysis that varied the cost parameter, the authors demonstrated how the approach smoothly switches to deletion-dominant techniques when additions become "expensive.”
- **Dependence on Attack Strategy (GcK9):** Ablation investigations demonstrated that attack-guided candidates greatly outperform random and heuristic baselines, allaying doubts regarding the necessity of the specific assault (GOttack).
- **Theoretical Claims (GcK9):** In order to solve the overclaiming issue, the authors reduced the language surrounding "Hypothesis 1" from a formal demonstration to a model-driven hypothesis supported by gradient reasoning.

**Outstanding Concerns:**

- **Domain-Specific Plausibility (GcK9):** The problem of these soft proxies remains, despite the authors' claims that their pruning techniques and structural priors (degree/motif) are sufficient for wide applications. True "plausibility" may require stringent bounds that the current loss function is unable to sufficiently represent in some domains (such as biology or finance). The authors did, however, recognize this as a path for further research.

**Reviewer Scores:**

- **Reviewer 7BBe:** **8**
- **Reviewer wuaS:** **6**
- **Reviewer GcK9:** **4**
- **Reviewer M2c7:** **6**

---

### Decision · Program_Chairs · 2026-01-26

Accept (Poster)